# Towards High Data Efficiency in Reinforcement Learning with Verifiable Reward

**Xinyu Tang**[1,3,*], **Zhenduo Zhang**[2,*], **Yurou Liu**[1,3,*], **Wayne Xin Zhao**[1,3,†],
**Zujie Wen**[2] , **Zhiqiang Zhang**[2] , **Jun Zhou**[2]
[1]Gaoling School of Artificial Intelligence, Renmin University of China [2]Ant Group
[3]Beijing Key Laboratory of Research on Large Models and Intelligent Governance

## Abstract

Recent advances in large language models (LLMs) have utilized reinforcement learning with verifiable rewards (RLVR) to improve reasoning capabilities. However, scaling these methods typically requires massive data and extensive rollout computations, leading to high training costs and low data efficiency. To mitigate this issue, we propose DEPO, a **D**ata-**E**fficient **P**olicy **O**ptimization approach that combines optimized strategies for both offline and online data selection. In the offline phase, we curate a high-quality subset of training data based on multiple objectives, including diversity, influence, and difficulty. During online RLVR training, we propose a sample-level explorability metric to dynamically filter out samples with low exploration potential, thereby reducing substantial rollout computational costs. Additionally, we employ a replay mechanism for under-explored samples to ensure sufficient training, which enhances the final convergence performance. Experiments on five reasoning benchmarks show that DEPO consistently outperforms existing methods in both offline and online data selection scenarios. Notably, using only **20%** of the training data, our approach achieves a **1.85** × speed-up on AIME24 and a **1.66** × speed-up on AIME25 compared to GRPO trained on the full dataset. The code is available at `https://github.com/RUCAIBox/DEPO`.

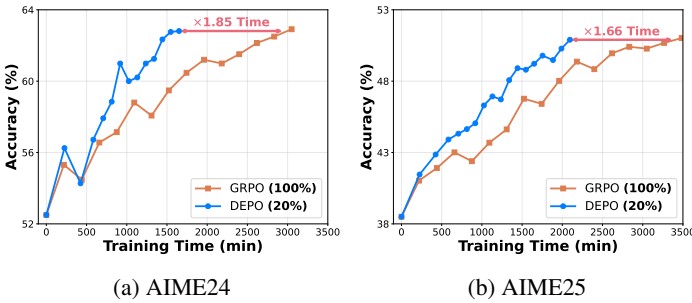

(a) AIME24    (b) AIME25

Figure 1: Training Accuracy of GRPO and DEPO. DEPO uses only 20% of the training data and reduces training time by at least 1.6 × while achieving comparable performance to GRPO.

## 1 Introduction

Recently, reinforcement learning with verifiable rewards (RLVR) (Ouyang et al., 2022; Shao et al., 2024; Tang et al., 2025b) has emerged as a prominent technique to unlock the reasoning capabilities of large language models (LLMs) (Jaech et al., 2024; Zhao et al., 2023a). In RLVR training, LLMs are required to explore multiple reasoning trajectories (*i.e.,* rollouts), and receive binary rewards based on the correctness of the final answer. This process enables LLMs to iteratively refine their reasoning strategies. A common way to enhance RLVR is to scale the amount of training data and the number of rollouts, which allows LLMs to discover more diverse reasoning paths and improve

---

* Equal contribution.
† Corresponding author.

performance. Despite its effectiveness, this scaling strategy introduces significant drawbacks, that is, *it substantially increases training cost and results in low **data efficiency***.

To mitigate this problem, prior work has explored ways to improve data efficiency through both offline and online data selection strategies. In offline settings, existing methods often rely on a single metric like reward trends (Li et al., 2025b), reward variance (Wang et al., 2025b), and gradient alignment (Li et al., 2025a) to select data, which fails to fully capture the complex characteristics of the training data. Furthermore, these approaches often require prior training to compute the metrics, which incurs high computational costs. On the other hand, online data selection methods aim to improve the rollout efficiency, which is the major bottleneck in RLVR. Zheng et al. (2025b) employs a probabilistic filter to exclude samples with historical zero reward variance. Although this method reduces rollout costs, it treats all historical non-zero variance samples equally and lacks a finer-grained metric to evaluate their exploration potential. Additionally, all existing methods enhance data efficiency from either the offline or the online perspective, resulting in suboptimal data efficiency.

In this work, we are the first to introduce a **D**ata-**E**fficient **P**olicy **O**ptimization approach that integrates optimized strategies for both offline and online data selection for RLVR, namely DEPO. During the offline phase, we apply a multi-objective high-quality data selection strategy. Specifically, to overcome the redundancy of training data, we propose a PageRank-weighted determinantal point process method to prune the dataset and preserve diverse and influential samples. Besides, to better align the difficulty of the dataset with the model's current capabilities, we perform offline rollouts on this pruned subset and select samples whose difficulty scores approximate a normal distribution. In the online RLVR training process, we tackle the computational inefficiency of exploring low-potential samples by proposing a sample-level explorability metric. This metric dynamically quantifies a sample's exploration potential based on its historical training dynamics, which allows us to strategically skip rollouts for low-explorability samples and allocate computational resources to samples with higher exploration potential for rollouts and policy updates. Moreover, to ensure all samples are adequately trained, we employ dynamic replay for under-explored samples to further improve the final convergence performance.

To validate the effectiveness and efficiency of our approach, we conduct experiments on five reasoning benchmarks. Experimental results show that DEPO outperforms several competitive baselines in both offline and online data selection settings. In particular, when using only **20%** of the training data, DEPO achieves a **1.85** times speed-up on AIME24 and a **1.66** times speed-up on AIME25 with DeepSeek-R1-Distill-Qwen-7B compared to GRPO trained on the full dataset. Our main contributions are summarized as follows:

- To the best of our knowledge, we are the first to integrate both offline and online data selection strategies to enhance data efficiency in RLVR training.

- In the offline phase, we employ a multi-dimensional data curation strategy based on diversity, influence, and difficulty. Then, during online training, we dynamically filter samples by their explorability and replay under-explored samples to further improve training efficiency.

- Extensive experiments across five reasoning datasets and three LLMs demonstrate the effectiveness and efficiency of our proposed method under both offline and online data selection scenarios.

## 2 RELATED WORK

Our work is related to Reinforcement Learning with Verifiable Reward (RLVR) and data efficiency. More detailed related work is presented in Appendix F.

**Reinforcement Learning with Verifiable Reward.** Reinforcement learning with verifiable reward (RLVR) effectively improves reasoning in LLMs, particularly for mathematics and code generation (Tang et al., 2025a; Zhan et al., 2025; Chen et al., 2026). It uses simple verification to provide binary rewards, eliminating the need for learned reward models. DeepSeek-R1 (DeepSeek-AI et al., 2025) first proposed the GRPO algorithm within this framework. Building on GRPO, subsequent work have further advanced RLVR by refining various aspects, including loss functions (Liu et al., 2025a; Yu et al., 2025; Zheng et al., 2025a; Chen et al., 2025a), token-level entropy (Wang et al., 2025a; Hao et al., 2025), advantage estimation (Cheng et al., 2025; Chen et al., 2025b), and hyperpa-

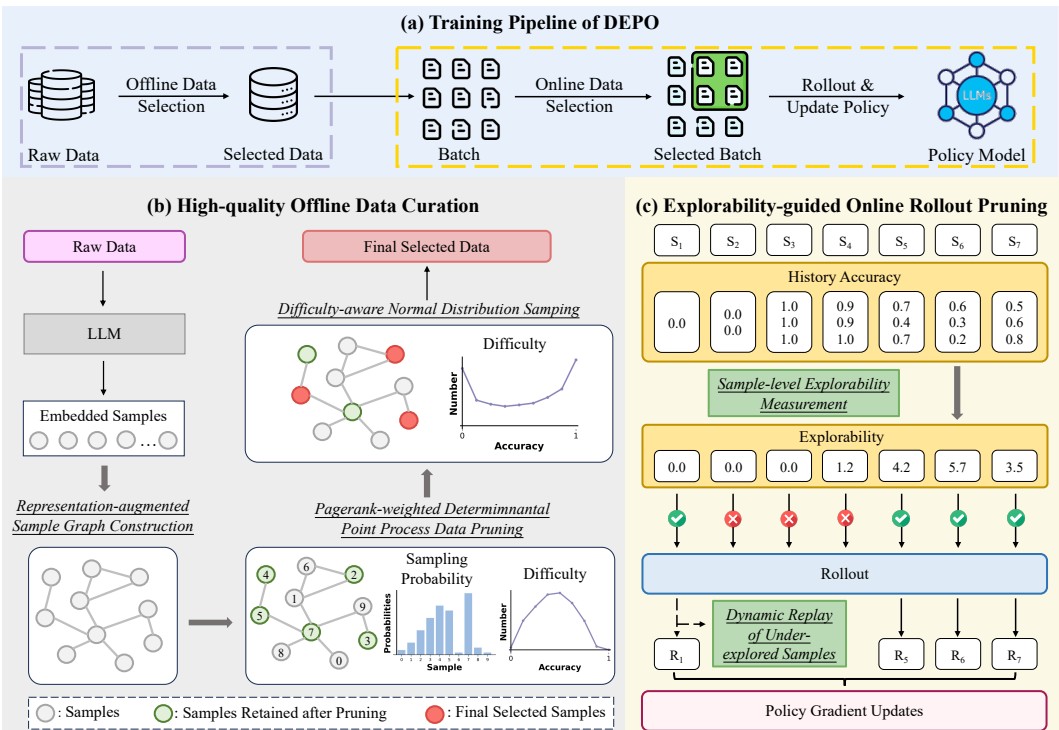

Figure 2: Overview of our approach DEPO. (a) Our approach improves the data efficiency in RLVR training via an end-to-end offline and online data selection strategy. (b) In the offline phase, we first construct a sample graph based on the representations, then apply PageRank-weighted Determinantal Point Process to select a diverse and influential subset, and finally sample from this subset with difficulty following a normal distribution. (c) In the online phase, we evaluate the explorability of each sample based on its historical training dynamics and retain high-explorability ones for rollout, and actively replay under-explored samples to ensure sufficient training of all samples.

rameter (Liu et al., 2025b; An et al., 2025; Xi et al., 2025). In this work, we focus on improving the data efficiency of RLVR to reduce computational costs while maintaining performance.

**Data Efficiency for RLVR.** Improving data efficiency in RLVR requires strategic selection of high-quality samples, incorporating offline and online data selection methods. Offline data selection methods focus on identifying a high-quality subset of data prior to training. Some studies select samples based on model reward trends (Li et al., 2025b), reward variance (Wang et al., 2025b), and gradient alignment (Li et al., 2025a). While effective, these methods require training the dataset for several epochs for selection. On the other hand, online data selection methods dynamically filter samples during the training process. GRESO (Zheng et al., 2025b) excludes samples with historical zero reward variance, but it lacks a finer-grained distinction among non-zero variance samples. In this paper, we integrate optimized offline and online selection to improve data efficiency for RLVR.

## 3 METHODOLOGY

In this section, we present DEPO, a method to enhance data efficiency in reinforcement learning with verifiable reward (RLVR). We first define the problem formulation, then describe a two-stage data selection process: (a) offline curation based on diversity, influence, and difficulty, and (b) online rollout pruning guided by explorability. Finally, we discuss the effectiveness and efficiency of our approach. The overall framework of DEPO is illustrated in Figure 2.

### 3.1 PROBLEM FORMULATION

In this work, we focus on a large language model (LLM) parameterized by $\theta \in \mathbb{R}^N$, which has been pretrained on large-scale corpora and will be further trained on an RLVR dataset $\mathcal{D} = \{x_i\}_{i=1}^{|\mathcal{D}|}$

to enhance its reasoning abilities. A key challenge in this process is the high computational cost associated with RLVR training, which typically requires extensive rollouts and policy updates over large datasets. To accelerate reasoning improvement, our goal is to optimize **data efficiency** (*i.e.,* achieve comparable performance with less training data and fewer rollouts) by leveraging $\mathcal{D}$ more efficiently without modification or augmentation. To achieve this, we first reduce data redundancy by selecting a high-quality subset offline. Then, to further speed up RLVR training, we select samples with high exploration potential during training. The two stages are detailed below.

## 3.2 Multi-dimensional Offline Data Curation

In this part, we present an offline data curation method to select a high-quality subset of RLVR data based on three criteria: diversity, influence, and difficulty. First, we construct a sample graph based on the representations. Next, we prune redundant samples by applying a PageRank-weighted Determinantal Point Process to retain a diverse and influential subset. Finally, we further refine this subset by selecting samples whose difficulty levels follow a normal distribution.

### 3.2.1 Diversity and Influence-aware Data Selection

**Representation-augmented Sample Graph Construction.** The first step of our approach is to model the relationships among samples using a graph. Previous studies (Hendel et al., 2023; Stolfo et al., 2025) demonstrate that internal model representations can effectively capture sample characteristics. Inspired by this, we follow Liu et al. (2024) and use the last token embedding from the final layer as the sample representation, as it aggregates the entire model information and input semantics. Based on these representations, we construct a sample graph $\mathcal{G} = (\mathbf{V}, \mathbf{E}, \mathbf{P})$, where each vertex $v_i \in \mathbf{V}$ denotes a sample, an edge $e(i, j) \in \mathbf{E}$ connects node $v_i$ and $v_j$, and edge weight matrix $\mathbf{P}$ encodes the pairwise similarities between their embeddings. This representation-augmented graph is subsequently used for our offline data selection process.

**Pagerank-weighted Determinantal Point Process Data Pruning.** We prune redundant data from the dataset by considering two properties: diversity and influence. Diversity ensures broad information coverage, while influence reflects the importance of samples in the graph. To promote diversity, we use Determinantal Point Process (DPP) (Kulesza & Taskar, 2012a) to identify a subset that maximizes the determinant of the corresponding similarity submatrix: $\max_{Y \subseteq \mathbf{P}} \det(Y)$, where $Y$ is the submatrix of the full similarity matrix $\mathbf{P}$, and $\det(\cdot)$ represents its determinant value. This refers to selecting samples that form a larger volume in the feature space, with a larger volume indicating greater diversity. To measure sample influence, we use PageRank (Brin & Page, 1998) to assign a weight $w_i \in \mathbf{w}$ to each sample, reflecting its representativeness. Our goal is to maximize the influence of the selected samples: $\max_{Y \subseteq \mathbf{P}} \prod_{i \in Y} w_i$. To unify both objectives, we combine diversity and influence into a single kernel matrix $L_Y$ and maximize its determinant:

$$\max_{Y \subseteq \mathbf{P}} \left( \det(Y) \cdot \prod_{i \in Y} w_i \right) = \max_{Y \subseteq \mathbf{P}} \det \left( \underbrace{\mathrm{diag}(\mathbf{w}_Y^{1/2}) \cdot Y \cdot \mathrm{diag}(\mathbf{w}_Y^{1/2})}_{L_Y} \right), \tag{1}$$

where $L_Y$ denotes the kernel matrix that integrates both diversity and influence of the selected subset. We provide detailed theoretical proof of this optimization objective in Appendix B.

Since the optimization problem is NP-hard, we follow Kulesza & Taskar (2012b) and employ a greedy algorithm to approximate the result. The algorithm iteratively selects samples based on a dynamically updated probability distribution and updates the probabilities of the remaining samples via Gram-Schmidt orthogonalization. The pseudo-code of this algorithm is presented in Algorithm 1. This process effectively reduces redundancy while retaining influential and diverse samples.

### 3.2.2 Difficulty-aware Normal Distribution Data Sampling

After pruning data via PageRank-weighted DPP, we obtain a representative and diverse subset $Y$. However, in RLVR training, samples that are too easy or too hard provide limited learning signals and contribute little to policy optimization. To better align the difficulty of training data with the

current model's capability, we propose a difficulty-aware sampling strategy that prioritizes samples of moderate difficulty. Specifically, for each sample $i$ in pruned subset $Y$, we generate $G$ offline trajectories using the current policy model $\pi_\theta$ and compute its accuracy score $\text{Acc}_i$:

$$\text{Acc}_i = \mathbb{E}_{\{o_j\}_{j=1}^G \sim \pi_\theta} \left[ \mathcal{V}(o_j, a_i) \right]. \tag{2}$$

Here, $\{o_j\}_{j=1}^G \sim \pi_\theta$ are $G$ responses generated by $\pi_\theta$ for question $i$, and $\mathcal{V}(o_j, a_i)$ denotes a verifier that evaluates whether model output $o_j$ matches the ground-truth answer $a_i$.

Using accuracy score as a measure of difficulty, we sample from $Y$ according to a normal distribution $\mathcal{N}(\mu, \sigma^2)$, where $\mu$ and $\sigma$ are the mean and standard deviation of the accuracies in the final selected subset $D^{\text{sub}}$. Thus, the sampling probability for each sample is proportional to the standard normal density function, which assigns higher probabilities to samples near the mean difficulty:

$$p_i = \frac{\phi\left(\frac{\text{Acc}_i - \mu}{\sigma}\right)}{\sum_{k \in Y} \phi\left(\frac{\text{Acc}_k - \mu}{\sigma}\right)}, \quad \text{where} \quad \phi(x) = \frac{1}{\sqrt{2\pi}} e^{-x^2/2}. \tag{3}$$

The final RL training subset $D^{\text{sub}}$ is constructed by sampling from $Y$ according to the probability distribution $\mathcal{P} = \{p_i\}_{i \in Y}$, which is subsequently used for RLVR training.

## 3.3 EXPLORABILITY-GUIDED ONLINE ROLLOUT PRUNING

In RLVR training, rollout generation is computationally expensive and becomes the main bottleneck for training speed. To further improve data efficiency, we propose an explorability metric to quantify the exploration potential of samples. Using this metric, we select high explorability samples for rollout generation and policy gradient updates and selectively skip unnecessary rollouts. Additionally, to avoid missing under-explored samples that may become valuable, we implement a dynamic replay mechanism to reuse them in the training process. The pseudo-code is presented in Algorithm 2.

### 3.3.1 SAMPLE-LEVEL EXPLORABILITY MEASUREMENT

In RLVR training, high-entropy samples encourage exploration, and low-entropy samples may lead to overfitting. Leveraging this effect, we define the explorability of a sample as the average entropy of its rollouts over a sliding window of recent epochs to evaluate its exploration potential. However, although high-entropy samples generally promote exploration, certain negative samples with excessively high entropy (*i.e.,* pathological trajectories) can introduce noise and destabilize training. To mitigate this, we apply a threshold $\lambda$ to exclude them. Specifically, we define the filtering function as:

$$I(q, a, o_i^t) = \mathbb{I}\left[ \mathcal{V}(o_i^t, a) = 1 \vee \left( \mathcal{V}(o_i^t, a) = 0 \wedge e(o_i^t) \leq \lambda e(\overline{o_i^{t+}}) \right) \right], \tag{4}$$

where $\mathbb{I}$ denotes the indicator function, $(q, a)$ is the question-answer pair, and $o_i^t$ is the $i$-th rollout in epoch $t$. The verification function $\mathcal{V}$ returns 1 if $o_i^t$ matches the ground truth $a$, and 0 otherwise. $e(o_i^t)$ is the average entropy across tokens, $e(\overline{o_i^{t+}})$ is the mean entropy of positive rollouts, and $\lambda$ serves as the threshold to filter high-entropy negative rollouts. We then compute the explorability of a single rollout $o_i^t$ by weighting its entropy by the absolute advantage $|\hat{A}_i|$ and the filtering indicator $I(q, a, o_i^t)$:

$$E(q, a, o_i^t) = |\hat{A}_i| \cdot e(o_i^t) \cdot I(q, a, o_i^t), \tag{5}$$

where $\hat{A}_i = \frac{r_i - \text{mean}(\{r_i\}_{i=1}^G)}{\text{std}(\{r_i\}_{i=1}^G)}$ is the advantage. This explorability metric prioritizes samples of moderate difficulty, which tend to maximize the sum of absolute advantages $\sum_{i=1}^G |\hat{A}_i|$. In contrast, samples with all rollouts either entirely correct or entirely incorrect yield zero advantage. Therefore, our explorability metric effectively captures both the **exploration potential** (entropy) and the **difficulty** (absolute advantage) of each sample. Finally, to evaluate a sample's exploration potential over time, we aggregate the explorability scores across the group over a sliding window of recent epochs to obtain the sample-level explorability $\mathcal{E}$.

$$\mathcal{E}(q, a, \{O^t\}_{t=e-s+1}^e) = \frac{1}{s} \sum_{t=e-s+1}^e \frac{1}{G} \sum_{i=1}^G E(q, a, o_i^t), \tag{6}$$

where $\{O^t\} = \{o_i^t\}_{i=1}^G$ are the rollouts in epoch $t$, and $s$ is the sliding window size of recent epochs.

Table 1: Computational time on 4×RTX 3090 GPUs.

| Component | Graph Construction | Pagerank-weighted DPP | Difficulty-aware Sampling |
|---|---|---|---|
| Time (h) | 0.75 (× 8.33) | 0.09 (× 1.00) | 44.33 (× 492.56) |

### 3.3.2 DYNAMIC REPLAY OF UNDER-EXPLORED SAMPLES

During training, samples with consistently low explorability may be overlooked, even if they become valuable in later stages. To mitigate this, we introduce a dynamic replay mechanism that proactively reintroduces under-explored samples. Specifically, each pruned batch $\mathcal{B}^{\text{Pruned}}$ consists of two types of samples: (1) the top $\alpha_e\%$ of samples from $\mathcal{B}$ ranked by their explorability $\mathcal{E}$, and (2) the bottom $\rho\%$ of samples from $\mathcal{B}$ ranked by their historical exploration frequency (*i.e.,* those selected the least number of times up to the current epoch $e$) Thus, the optimization objective for DEPO is:

$$
\mathcal{J}_{\text{DEPO}}(\theta) = \mathbb{E}_{\mathcal{B}\sim\mathcal{D}^{\text{sub}},(q,a)\sim\mathcal{B},\{o_i\}_{i=1}^G\sim\pi_{\theta_{\text{old}}}(\cdot|q)} \left[ \mathbb{I}\Big[q,a,\{O^t\}_{t=e-s+1}^e\Big] \frac{1}{G}\sum_{i=1}^G \cdot \frac{1}{|o_i|}\sum_{t=1}^{|o_i|} \right.
$$
$$
\left. \left( \min\left(r_{i,t}(\theta)\hat{A}_i,\ \text{clip}\left(r_{i,t}(\theta), 1-\varepsilon, 1+\varepsilon\right)\hat{A}_i\right) - \beta D_{\text{KL}}(\pi_\theta||\pi_{\text{ref}}) \right) \right], \tag{7}
$$

$$
\text{where} \quad \mathbb{I}\Big[q,a,\{O^t\}_{t=e-s+1}^e\Big] = \begin{cases} 1 & \mathcal{E}(q,a,\{O^t\}_{t=e-s+1}^e) \text{ is top-}\alpha_e\% \\ 1 & |\{O^t\}_{t=1}^e| \text{ is bottom-}\rho\% \\ 0 & \text{else,} \end{cases}
$$

$$
r_{i,t}(\theta) = \frac{\pi_\theta(o_{i,t} \mid q, o_{i,<t})}{\pi_{\theta_{\text{old}}}(o_{i,t} \mid q, o_{i,<t})}, \quad \text{and} \quad \alpha_e = \alpha_0 - d \cdot e. \tag{8}
$$

To adaptively shift the training focus from broad exploration in the early stages to specialized refinement in later phases, we dynamically reduce the proportion of high-explorability samples per epoch using a linear decay rate $d$. Here, $\mathcal{B}$ is the samples in a training batch from the offline selected subset $\mathcal{D}^{\text{sub}}$, $\alpha_0$ denotes the initial sampling rate, $d$ is the decay rate, and $e$ is the current epoch. The online rollout pruning strategy optimizes computational resource allocation by focusing on samples with high exploration potential and strategically replaying under-explored samples to ensure all samples are sufficiently trained, significantly improving training efficiency.

### 3.4 DISCUSSION

**Effectiveness of DEPO.** DEPO enhances the data efficiency of RLVR via both offline and online data selection methods. During the offline phase, DEPO selects high-quality subsets by jointly optimizing three criteria: diversity, influence, and difficulty, which overcome the problem of prior methods that rely on single metrics such as reward trends (Li et al., 2025b), reward variance (Wang et al., 2025b), and gradient alignment (Li et al., 2025a). Besides, these metrics rely only on early-stage training dynamics, which limits their effectiveness in later training phases. In the online phase, DEPO dynamically prioritizes samples based on their explorability, which is a fine-grained measure of exploration potential. In contrast, GRESO (Zheng et al., 2025b) probabilistically removes samples with historical zero-variance rewards, which treats all historical non-zero variance samples equally and underperforms when zero variance samples are scarce.

**Efficiency of DEPO.** DEPO improves data efficiency by jointly optimizing offline and online data selection, thereby reducing both the amount of training data and the rollout numbers. In contrast, existing methods typically address data efficiency from only one perspective (*i.e.,* either offline or online). In the offline phase, DEPO employs a two-step strategy to minimize computational overhead, which first prunes the dataset using a PageRank-weighted DPP, and then performs offline rollouts only on this reduced subset.

This method is more efficient than performing rollouts over the full dataset (An et al., 2025), as the DPP-based filtering significantly reduces the dataset size, thereby lowering the cost of offline

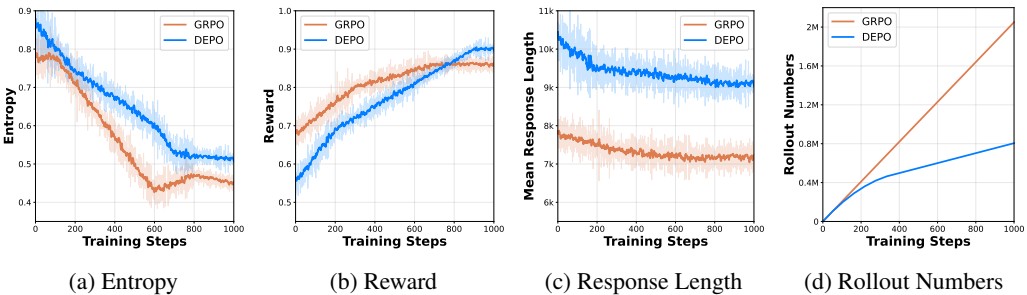

(a) Entropy      (b) Reward      (c) Response Length      (d) Rollout Numbers

Figure 3: The RLVR training dynamics of GRPO and DEPO.

rollouts. As shown in Table 1, difficulty sampling accounts for a significantly larger portion of the total time compared to other components. Moreover, unlike offline methods that require multiple epochs of training on the original or a warm-up dataset to guide selection (Li et al., 2025b; Wang et al., 2025b; Li et al., 2025a), our approach avoids such additional training costs. In the online phase, DEPO dynamically estimates sample explorability based on historical training dynamics, which helps minimize rollout costs and improve training efficiency. In contrast, some methods Yu et al. (2025) rely on extensive rollouts before each update, incurring significant computational costs.

## 4 EXPERIMENTS

In this section, we first describe the experimental setup, then present the main results and provide a detailed analysis.

### 4.1 EXPERIMENTAL SETUP

We run experiments on DeepSeek-R1-Distill-Qwen-7B, DeepSeek-R1-Distill-Llama-8B (DeepSeek-AI et al., 2025), Qwen2.5-Math-7B (Yang et al., 2024). We use DAPO-Math (Yu et al., 2025) as the training dataset and apply the GRPO algorithm to train the models. For evaluation benchmarks, we use three mathematical reasoning benchmarks (*i.e.,* AIME24, AIME25, Math500 (Hendrycks et al., 2021)) and two other reasoning benchmarks (*i.e.,* GPQA (Rein et al., 2023) and LiveCodeBench (Jain et al., 2025)). We follow Zheng et al. (2025b) to evaluate models every 50 steps and report the performance that achieves the best average performance across five benchmarks. We repeat the test set 32 times for all benchmarks and report the average accuracy. More experimental details are provided in Appendix C. To enable a systematic comparison, we include several representative offline and online data selection methods as baselines. For offline selection methods, we compare our method with random selection, supervised fine-tuning (SFT)-based methods (*i.e.,* PPL-Top (Laurençon et al., 2022) and PPL-Middle (Ankner et al., 2025)), and RLVR selection methods (*i.e.,* LIMR (Li et al., 2025b) and Learnalign (Li et al., 2025a)). For online selection methods, we integrate them into our offline selected subset and compare DEPO with random online selection and GRESO (Zheng et al., 2025b). Detailed descriptions of these baselines are provided in Appendix E.

### 4.2 MAIN RESULTS

In this section, we present the main experimental results. Figure 3 illustrates the training dynamics of DEPO and GRPO, and Table 2 presents the main results.

**DEPO selects high-quality subsets offline for RLVR training.** As shown in Table 2, reinforcement learning on mathematical data not only enhances mathematical reasoning but also improves the performance on other reasoning tasks. Among offline data selection methods, although SFT-based data selection methods (*i.e.,* PPL-Top and PPL-Middle) incorporate additional information from training data, their performance performs poorly. This may stem from the mismatch between SFT and RL objectives. SFT maximizes the likelihood of target outputs, making perplexity a natural indicator of sample difficulty. In contrast, RL aims to maximize rewards, requiring samples to match the model's current capability. Moreover, RLVR methods (*i.e.,* LIMR and Learnalign), which perform training before selection, lead to further improvements. However, these approaches tend to select

Table 2: Performance comparison of various data selection methods. "**Offline**" and "**Online**" refer to the offline and online data selection methods, respectively. "**Ratio**, "**Time**", and "**RN**" denote the ratio of selected data, total training time, and total rollout numbers, respectively. We highlight the best performance across different data selection methods. Numbers marked with * indicate that the improvement is statistically significant compared with baselines (t-test with p-value $< 0.05$).

| Model | Method | | Accuracy | | | | | | Efficiency | | |
|---|---|---|---|---|---|---|---|---|---|---|---|
| | | | AIME 24 | AIME 25 | MATH500 | GPQA | LiveCodeBench | Average | Ratio | Time | RN |
| **Deepseek-R1-Distill-Qwen-7B** | - | Base | $51.5_{\pm1.0}$ | $38.5_{\pm0.7}$ | $91.2_{\pm1.1}$ | $45.9_{\pm1.4}$ | $37.0_{\pm0.7}$ | $52.8_{\pm1.0}$ | - | - | - |
| | | Full | $63.4_{\pm1.4}$ | $52.7_{\pm0.9}$ | $96.3_{\pm1.1}$ | $51.6_{\pm1.4}$ | $44.7_{\pm1.4}$ | $61.7_{\pm1.2}$ | 100% | 100% | 100% |
| | Offline | Random | $56.8_{\pm0.9}$ | $43.0_{\pm0.7}$ | $94.6_{\pm1.2}$ | $47.3_{\pm0.6}$ | $39.6_{\pm1.4}$ | $56.3_{\pm1.0}$ | 20% | 98% | 100% |
| | | PPL-Top | $57.5_{\pm0.6}$ | $45.3_{\pm0.7}$ | $95.0_{\pm0.2}$ | $48.2_{\pm0.2}$ | $40.4_{\pm0.7}$ | $57.3_{\pm0.5}$ | 20% | 101% | 100% |
| | | PPL-Middle | $57.5_{\pm0.2}$ | $45.4_{\pm0.9}$ | $95.2_{\pm0.7}$ | $48.6_{\pm0.5}$ | $40.8_{\pm0.5}$ | $57.5_{\pm0.6}$ | 20% | 97% | 100% |
| | | LIMR | $59.9_{\pm0.8}$ | $46.4_{\pm0.7}$ | $95.1_{\pm1.2}$ | $48.5_{\pm0.7}$ | $40.9_{\pm0.8}$ | $58.2_{\pm0.9}$ | 20% | 99% | 100% |
| | | Learnalign | $60.1_{\pm0.9}$ | $46.8_{\pm0.9}$ | $95.5_{\pm0.4}$ | $49.0_{\pm0.4}$ | $41.9_{\pm0.2}$ | $58.7_{\pm0.6}$ | 20% | 102% | 100% |
| | | **DEPO-Offline** | $\mathbf{63.1^*_{\pm1.2}}$ | $\mathbf{51.7^*_{\pm1.1}}$ | $\mathbf{96.1^*_{\pm0.2}}$ | $\mathbf{51.7^*_{\pm1.1}}$ | $\mathbf{44.5^*_{\pm0.2}}$ | $\mathbf{61.4^*_{\pm0.8}}$ | 20% | 99% | 100% |
| | Online | + Random | $58.7_{\pm0.8}$ | $45.3_{\pm0.5}$ | $93.1_{\pm0.6}$ | $47.2_{\pm1.3}$ | $39.3_{\pm0.7}$ | $56.7_{\pm0.8}$ | 20% | 58% | 40% |
| | | + GRESO | $60.2_{\pm0.2}$ | $47.4_{\pm1.0}$ | $94.3_{\pm0.7}$ | $48.1_{\pm0.4}$ | $40.6_{\pm0.9}$ | $58.1_{\pm0.6}$ | 20% | 55% | 40% |
| | | + DEPO | $\mathbf{62.8^*_{\pm0.4}}$ | $\mathbf{50.9^*_{\pm1.0}}$ | $\mathbf{95.9^*_{\pm0.4}}$ | $\mathbf{51.4^*_{\pm0.5}}$ | $\mathbf{44.3^*_{\pm0.5}}$ | $\mathbf{61.1^*_{\pm0.6}}$ | 20% | 57% | 40% |
| **Deepseek-R1-Distill-Llama-8B** | - | Base | $41.1_{\pm1.0}$ | $30.4_{\pm0.5}$ | $88.5_{\pm1.3}$ | $37.3_{\pm0.6}$ | $44.3_{\pm1.0}$ | $48.3_{\pm0.9}$ | - | - | - |
| | | Full | $56.9_{\pm0.8}$ | $45.1_{\pm0.9}$ | $94.8_{\pm0.7}$ | $44.4_{\pm0.4}$ | $49.6_{\pm0.6}$ | $58.2_{\pm0.7}$ | 100% | 100% | 100% |
| | Offline | Random | $47.6_{\pm0.4}$ | $38.7_{\pm0.9}$ | $90.6_{\pm1.0}$ | $39.6_{\pm0.5}$ | $44.5_{\pm0.6}$ | $52.2_{\pm0.7}$ | 20% | 100% | 100% |
| | | PPL-Top | $48.2_{\pm0.7}$ | $39.3_{\pm0.8}$ | $90.4_{\pm1.0}$ | $39.1_{\pm0.9}$ | $45.0_{\pm1.1}$ | $52.4_{\pm0.9}$ | 20% | 102% | 100% |
| | | PPL-Middle | $49.9_{\pm0.3}$ | $39.2_{\pm1.1}$ | $91.1_{\pm0.9}$ | $39.4_{\pm0.9}$ | $45.7_{\pm0.7}$ | $53.1_{\pm0.8}$ | 20% | 102% | 100% |
| | | LIMR | $52.3_{\pm1.2}$ | $40.9_{\pm0.3}$ | $91.6_{\pm1.0}$ | $41.0_{\pm1.0}$ | $45.3_{\pm1.3}$ | $54.2_{\pm1.0}$ | 20% | 97% | 100% |
| | | Learnalign | $54.7_{\pm1.2}$ | $41.8_{\pm1.3}$ | $91.6_{\pm0.7}$ | $40.8_{\pm1.0}$ | $46.2_{\pm0.9}$ | $55.0_{\pm1.0}$ | 20% | 98% | 100% |
| | | **DEPO-Offline** | $\mathbf{57.6^*_{\pm0.5}}$ | $\mathbf{44.8^*_{\pm1.1}}$ | $\mathbf{94.2^*_{\pm0.6}}$ | $\mathbf{43.6^*_{\pm1.1}}$ | $\mathbf{49.3^*_{\pm0.6}}$ | $\mathbf{57.9^*_{\pm0.8}}$ | 20% | 100% | 100% |
| | Online | + Random | $50.2_{\pm0.3}$ | $38.4_{\pm0.9}$ | $90.1_{\pm1.2}$ | $39.7_{\pm0.5}$ | $44.8_{\pm0.9}$ | $52.6_{\pm0.8}$ | 20% | 55% | 40% |
| | | + GRESO | $52.6_{\pm0.6}$ | $40.2_{\pm0.4}$ | $92.0_{\pm0.9}$ | $40.5_{\pm0.7}$ | $46.6_{\pm1.0}$ | $54.4_{\pm0.7}$ | 20% | 54% | 40% |
| | | + DEPO | $\mathbf{56.8^*_{\pm0.9}}$ | $\mathbf{44.4^*_{\pm1.1}}$ | $\mathbf{93.7^*_{\pm0.5}}$ | $\mathbf{42.8^*_{\pm0.5}}$ | $\mathbf{48.8^*_{\pm0.5}}$ | $\mathbf{57.3^*_{\pm0.7}}$ | 20% | 56% | 40% |
| **Qwen2.5-7B-Math** | - | Base | $13.4_{\pm0.8}$ | $6.4_{\pm0.7}$ | $54.5_{\pm1.1}$ | $28.7_{\pm0.3}$ | $5.6_{\pm0.6}$ | $21.7_{\pm0.7}$ | - | - | - |
| | | Full | $30.2_{\pm0.4}$ | $20.3_{\pm1.1}$ | $86.8_{\pm0.8}$ | $35.7_{\pm0.7}$ | $13.6_{\pm0.5}$ | $37.3_{\pm0.7}$ | 100% | 100% | 100% |
| | Offline | Random | $22.5_{\pm0.7}$ | $13.3_{\pm1.1}$ | $72.5_{\pm0.5}$ | $30.3_{\pm0.3}$ | $8.2_{\pm0.6}$ | $29.4_{\pm0.6}$ | 20% | 98% | 100% |
| | | PPL-Top | $24.1_{\pm1.0}$ | $13.8_{\pm0.9}$ | $76.2_{\pm1.0}$ | $31.0_{\pm0.6}$ | $9.6_{\pm0.6}$ | $30.9_{\pm0.8}$ | 20% | 102% | 100% |
| | | PPL-Middle | $24.8_{\pm0.7}$ | $14.3_{\pm1.1}$ | $76.0_{\pm0.8}$ | $30.6_{\pm0.7}$ | $9.9_{\pm0.8}$ | $31.1_{\pm0.8}$ | 20% | 98% | 100% |
| | | LIMR | $26.5_{\pm0.3}$ | $15.8_{\pm0.8}$ | $78.0_{\pm0.6}$ | $32.2_{\pm1.1}$ | $10.6_{\pm0.7}$ | $32.6_{\pm0.7}$ | 20% | 101% | 100% |
| | | Learnalign | $27.1_{\pm1.0}$ | $17.2_{\pm1.1}$ | $80.5_{\pm0.5}$ | $33.6_{\pm1.1}$ | $10.9_{\pm0.7}$ | $33.9_{\pm0.9}$ | 20% | 98% | 100% |
| | | **DEPO-Offline** | $\mathbf{30.0^*_{\pm0.7}}$ | $\mathbf{19.4^*_{\pm0.3}}$ | $\mathbf{85.8^*_{\pm0.4}}$ | $\mathbf{35.2^*_{\pm0.3}}$ | $\mathbf{13.2^*_{\pm0.3}}$ | $\mathbf{36.7^*_{\pm0.4}}$ | 20% | 99% | 100% |
| | Online | + Random | $24.3_{\pm0.7}$ | $14.5_{\pm1.1}$ | $74.3_{\pm1.1}$ | $31.1_{\pm0.3}$ | $9.3_{\pm0.8}$ | $30.7_{\pm0.8}$ | 20% | 57% | 40% |
| | | + GRESO | $27.7_{\pm0.3}$ | $16.8_{\pm0.5}$ | $80.7_{\pm0.9}$ | $33.4_{\pm0.9}$ | $10.9_{\pm0.4}$ | $33.9_{\pm0.6}$ | 20% | 57% | 40% |
| | | + DEPO | $\mathbf{29.8^*_{\pm0.3}}$ | $\mathbf{19.2^*_{\pm0.6}}$ | $\mathbf{86.3^*_{\pm0.6}}$ | $\mathbf{34.8^*_{\pm1.0}}$ | $\mathbf{12.8^*_{\pm0.7}}$ | $\mathbf{36.6^*_{\pm0.6}}$ | 20% | 55% | 40% |

samples that match the initial model capabilities, resulting in improvements during early stages but limiting performance in later phases. Additionally, they often overlook interdependencies among problems. In contrast, DEPO achieves the best performance across all methods, nearly matching the performance of training on the full dataset. One possible reason is that DEPO selects samples with diversity, influence, and appropriate difficulty, ensuring rapid improvement during the early stage. Besides, the diversity in data difficulty supports sustained improvement in later stages. As illustrated in Figure 3, our method selects samples with higher initial entropy, lower initial rewards, and longer response lengths. This ensures that our selected data matches the model's current capability and offers diverse exploration paths for effective RLVR training.

**DEPO saves training computational costs and maintains comparable performance.** As shown in Table 2, randomly reducing rollouts severely degrades performance, which highlights the importance of careful online sample selection. Moreover, GRESO improves model performance by filtering out historical zero-variance samples that contribute little to training. However, it does not account for the differences among historical non-zero-variance samples, which limits its overall performance. In comparison, DEPO achieves performance comparable to full rollouts while using less than 60% of the training time and 40% of the rollout budgets. This is because DEPO dynamically estimates sample explorability based on historical training dynamics, prioritizing highly explorable samples for rollouts and policy updates. Furthermore, we incorporate a replay strategy for under-explored samples to ensure sufficient training across all data, which leads to better final convergence performance. As illustrated in Figure 3, our rollout pruning strategy consistently selects samples with higher entropy, faster reward growth, and longer responses throughout the training process, demonstrating that DEPO efficiently improves reasoning performance.

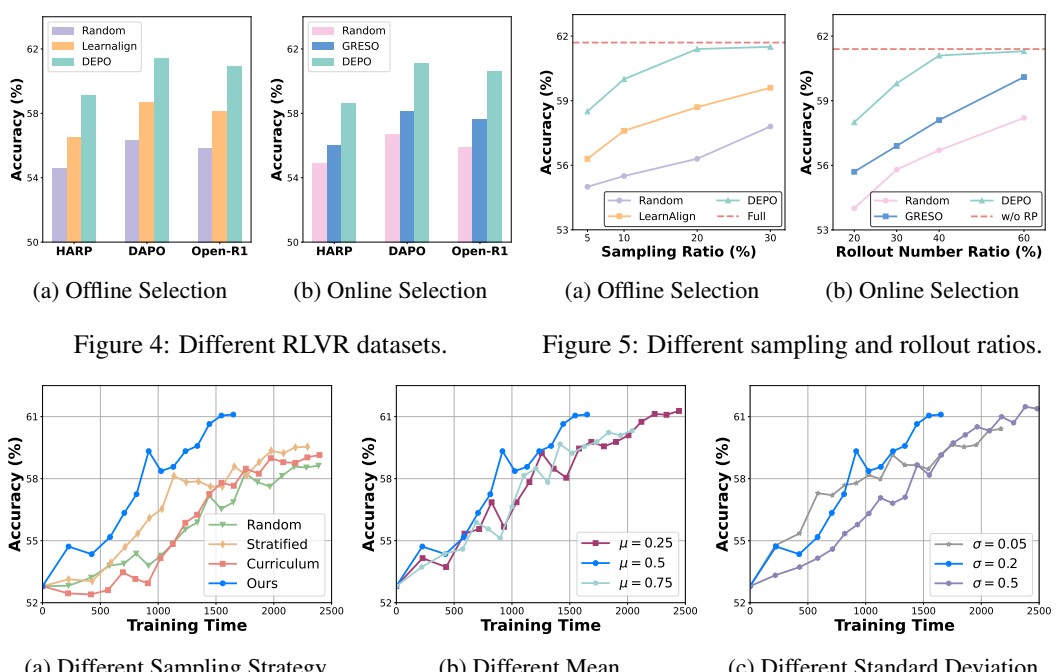

Figure 4: Different RLVR datasets.

Figure 5: Different sampling and rollout ratios.

(a) Different Sampling Strategy    (b) Different Mean    (c) Different Standard Deviation

Figure 6: Performance comparison across data of different difficulty levels.

## 4.3 DETAILED ANALYSIS

In this part, we conduct a detailed analysis of our proposed method. Unless stated otherwise, we report the average accuracy across five benchmarks using DeepSeek-R1-Distill-Qwen-7B. We provide additional detailed analysis in Appendix D.

**DEPO performs well using different RLVR training datasets.** We conduct experiments on three datasets of varying sizes (*i.e.,* HARP (Yue et al., 2024) (5k samples), DAPO (Yu et al., 2025) (17k samples), and Open-R1 (Face, 2025) (30k samples)). As shown in Figure 4, we observe that training on the DAPO dataset yields the best performance, indicating that higher data quality is more beneficial than larger data quantity in the RLVR training process. Moreover, DEPO outperforms competitive baselines in both offline and online settings across all datasets. These results confirm that DEPO is capable of improving data efficiency with different volumes of data.

**DEPO performs well under different offline data sampling ratios.** Figure 5a compares offline data selection methods under varying sampling ratios. As we can see, DEPO consistently outperforms Random and LearnAlign across all ratios. Notably, using only 20% of the data, DEPO matches the performance of full-dataset training. This indicates that our approach identifies high-value samples, allowing the model to efficiently improve its reasoning capabilities with limited data. Moreover, we observe that the performance of our method initially improves with increased sampling ratios, and it plateaus at around 20%. This also suggests that the dataset contains a substantial portion of redundant and low-value samples, which contribute little to the model's reasoning performance.

**DEPO achieves the best performance across different rollout numbers.** Figure 5b presents the performance of online data selection methods under different rollout ratios. As we can see, our proposed method consistently outperforms other online data selection baselines under different rollout budgets. This demonstrates that selectively performing rollouts on samples with high explorability significantly improves training efficiency without sacrificing performance. These results show the effectiveness of using explorability as a metric to select samples in the RLVR process.

**Normal distribution sampling aligns dataset difficulty with model capability.** To evaluate the impact of different sampling strategies, we compare our approach with random, stratified sampling, and an easy-to-hard curriculum learning strategy. Random sampling preserves the original U-

shaped difficulty distribution, with many samples being entirely correct or incorrect. Stratified sampling selects equal proportions from each difficulty level, and the easy-to-hard curriculum strategy progressively increases the difficulty of training samples over time. As shown in Figure 6a, random sampling performs worst, followed by the easy-to-hard curriculum and stratified sampling, and our method achieves the best results. The easy-to-hard curriculum strategy leads to slow initial progress because overly simple samples provide limited learning signals. Its performance improves noticeably in the middle phase when moderately difficult samples are introduced, but the model fails to achieve higher final accuracy, as overly difficult samples also contribute little to learning. These results confirm that extremely easy or hard samples offer limited training signals, but DEPO prioritizes moderately difficult samples that are more beneficial.

**Medium-difficulty samples accelerate learning, while the inclusion of challenging samples improves final convergence performance.** To further analyze the effect of difficulty-aware normal distribution sampling, we vary the mean and standard deviation of the sampling distribution in Figure 6b and Figure 6c. Results indicate that relatively easier samples (*i.e.,* $\mu = 0.75$) lead to lower convergence performance, while harder ones (*i.e.,* $\mu = 0.25$) improve final results but learn slower. Additionally, a smaller standard deviation (*i.e.,* $\sigma = 0.05$) speeds up early learning but limits the final performance, whereas a larger standard deviation (*i.e.,* $\sigma = 0.5$) slows initial progress but achieves higher convergence. These findings confirm that medium-difficulty samples facilitate rapid improvement, and incorporating some challenging samples is essential for the final peak performance.

## 4.4 ABLATION STUDIES

To evaluate the effectiveness of each component in our method, we conduct ablation studies on three math benchmarks using DeepSeek-R1-Distill-Qwen-7B. As shown in Table 3, removing any component leads to performance degradation, demonstrating all components are essential. In offline selection, removing Difficulty-aware Sampling leads to the most significant drop, which indicates that sample difficulty is a crucial factor for selection. For online selection, replacing explorability-based filtering with random

Table 3: Ablation study on three math benchmarks.

| Dataset | AIME 24 | AIME 25 | MATH500 |
|---|---|---|---|
| DEPO | **62.8** | **50.9** | **95.9** |
| *Offline Data Selection* | | | |
| w/o Pagerank-weighted DPP | 62.1 | 50.0 | 95.6 |
| w/o Difficulty-aware Sampling | 60.3 | 47.8 | 95.1 |
| *Online Data Selection* | | | |
| w/o Explorability Measurement | 58.7 | 45.3 | 93.1 |
| w/o Absolute advantage | 61.9 | 49.5 | 95.5 |
| w/o Entropy | 60.6 | 48.4 | 94.6 |
| w/o Under-explored Sample Replay | 62.3 | 48.4 | 95.2 |

filtering substantially reduces performance. This confirms that explorability is an important indicator of a sample's value in RLVR training. Furthermore, removing Under-explored Sample Replay notably impairs performance on more challenging tasks such as AIME 25 (performance drops from 50.9 to 48.4), suggesting that replaying challenging and under-trained samples is critical for enhancing the model's ability to solve hard problems.

## 5 CONCLUSION

In this paper, we proposed DEPO, a data-efficient policy optimization pipeline that integrates offline and online data selection strategies. By first constructing a high-quality subset of training data that emphasizes diversity, influence, and appropriate difficulty, then dynamically filtering rollouts based on sample-level explorability, our approach significantly reduced both data volume and computational costs while maintaining strong performance. Extensive experiments across multiple reasoning benchmarks and LLMs demonstrate that DEPO consistently outperformed competitive baselines in both offline and online settings. We hope our work inspires future research toward developing more data-efficient methods to accelerate RL for LLMs.

## ACKNOWLEDGMENTS

This work was partially supported by the National Natural Science Foundation of China No. 92470205 and Beijing Major Science and Technology Project under Contract No. Z251100008425002. This work was also supported by Ant Group Research Fund. Xin Zhao is the corresponding author.

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

---

**Algorithm 1** Pagerank-weighted Sequential Data Pruning$(S, \mathbf{w}, n.k)$

---

**Inputs:**
    Similarity matrix $S \in \mathbb{R}^{n \times n}$, Pagerank-weighted vector $\mathbf{w} \in \mathbb{R}^n$, Total node $n$, Sample size $k \leq n$
**Initialize:**
    Kernel Matrix $L = \text{diag}(\mathbf{w}^{1/2}) \cdot S \cdot \text{diag}(\mathbf{w}^{1/2})$, $Y \leftarrow \emptyset$, $\mathcal{C} \leftarrow \{1, 2, \ldots, n\}$, $L = Q\Lambda Q^\top$,
    $\Lambda \leftarrow \text{diag}(\lambda_1, \ldots, \lambda_n)$, $V \leftarrow Q \cdot \text{diag}(\lambda_1^{-1/2}, \ldots, \lambda_n^{-1/2})$
**for** $t = 1$ **to** $k$ **do**
    $\mathbf{p} \leftarrow \mathbf{0}_{|\mathcal{C}|}$                                        $\triangleright$ Initialize zero vector: $\mathbf{p} \in \mathbb{R}^{|\mathcal{C}|}, p_i = 0\ \forall i$
    **for** $i = 1$ **to** $n$ **do**
        **if** $i \in \mathcal{C}$ **then**
            $\mathbf{p}[i] \leftarrow \|V_i\|_2^2$
        **end if**
    **end for**
    $\mathbf{p} \leftarrow \mathbf{p}/\sum_j \mathbf{p}[j]$
    $i_t \leftarrow \text{Sample}(\mathcal{C}, \mathbf{p})$
    $Y \leftarrow Y \cup \{i_t\}$
    $\mathcal{C} \leftarrow \mathcal{C} \setminus \{i_t\}$
    **if** $t < k$ **then**
        $\mathbf{u} \leftarrow V_{i_t}^\top$                                     $\triangleright$ Select pivot vector
        $\mathbf{c} \leftarrow \mathbf{u}^\top V_{\mathcal{C}}$
        $V_{\mathcal{C}} \leftarrow V_{\mathcal{C}} - \mathbf{u}\mathbf{c}$                   $\triangleright$ Gram-Schmidt orthogonalization
    **end if**
**end for**
**return** $Y$

---

**Algorithm 2** Explorability-guided Rollout Pruning$(\mathcal{B}, s, \lambda, \rho, d, e, G, \{O^t\}_{t=1}^e)$

---

**Inputs:**
    Raw batch $\mathcal{B} = (q_i, a_i)_{i=1}^{|\mathcal{B}|}$, Window size $s$, Threshold for filtering poor negative rollouts $\lambda$, Replay ratio $\rho$,
    Decay rate $d$, Current epoch number $e$, Rollout numbers per sample $G$, Rollout history $\{O^t\}_{t=1}^e$
**Initialize:**
    Pruned batch $\mathcal{B}^{\text{Pruned}} \leftarrow \emptyset$
**for** each sample $(q_i, a_i)$ in $\mathcal{B}$ **do**
    $\mathcal{E}_i \leftarrow \mathcal{E}(q_i, a_i, \{O_i^t\}_{t=e-s+1}^e)$          $\triangleright$ Calculate sample-level explorability using Equation **??**
**end for**
$\alpha_e = \alpha_0 - d \cdot e$                    $\triangleright$ Calculate high explorability ratios for this epoch
Sort $\mathcal{B}$ in descending order by $\mathcal{E}_i$
$\mathcal{B}^{\text{High-Exp}} \leftarrow$ top $\lceil \alpha_e \times |\mathcal{B}| \rceil$ samples from sorted $\mathcal{B}$
$\mathcal{B}^{\text{Replay}} \leftarrow$ sample $\lceil \rho \times |\mathcal{B}| \rceil$ samples from $\mathcal{B}$ with the smallest $|\{O^t\}_{t=1}^e|$
$\mathcal{B}^{\text{Pruned}} \leftarrow \mathcal{B}^{\text{High-Exp}} \cup \mathcal{B}^{\text{Replay}}$
**return** $\mathcal{B}^{\text{Pruned}}$

---

# A  PSEUDO CODE

In this section, we present the detailed pseudo code for Pagerank-weighted sequential data pruning in Algorithm 1 and explorability-guided rollout pruning in Algorithm 2.

# B  THEORETIC PROOF

In this part, we provide a theoretical derivation of the optimization objective aimed at maximizing the determinant of a kernel matrix that incorporates both diversity and influence in Equation 1.

## B.1  OPTIMIZATION OBJECTIVE REFORMULATION

We begin with the original weighted determinant maximization problem:

$$\max_{Y \subseteq \mathbf{P}} \left( \det(S_Y) \cdot \prod_{i \in Y} w_i \right), \tag{9}$$

where $S_Y \in \mathbb{R}^{|Y| \times |Y|}$ denotes the similarity matrix over the subset $Y$, and $w_i \in \mathbf{w}$ represents the influential weight (*i.e.,* the PageRank score) of each sample $i$. This objective aims to select a subset $Y$ that is both diverse (as captured by $\det(S_Y)$) and influential (as promoted by the product of weights $\prod_{i \in Y} w_i$). To combine these two factors more naturally, we express the product of weights in matrix form. Note that:

$$\prod_{i \in Y} w_i = \det(\text{diag}(w_Y)), \tag{10}$$

where $\text{diag}(w_Y)$ is the diagonal matrix formed by the weights $w_i$. To incorporate the weights directly into the kernel matrix, we consider the square root of the weights.

$$\text{diag}(w_Y^{1/2}) = \text{diag}(\sqrt{w_i})_{i \in Y}. \tag{11}$$

We then observe that:

$$\max_{Y \subseteq \mathbf{P}} \left( \det(S_Y) \cdot \prod_{i \in Y} w_i \right) = \max_{Y \subseteq \mathbf{P}} \det\left( \text{diag}(\mathbf{w}_Y^{1/2}) \cdot S_Y \cdot \text{diag}(\mathbf{w}_Y^{1/2}) \right). \tag{12}$$

This equality follows from the multiplicative property of the determinant and the fact that $\text{diag}(\mathbf{w}_Y^{1/2})$ is a diagonal matrix. Now, defining the weighted kernel matrix for the subset $Y$ as:

$$L_Y = \text{diag}(\mathbf{w}_Y^{1/2}) \cdot S_Y \cdot \text{diag}(\mathbf{w}_Y^{1/2}), \tag{13}$$

we can rewrite the optimization problem as:

$$\max_{Y \subseteq \mathbf{P}} \det(L_Y) \tag{14}$$

This form corresponds to a standard determinantal point process (DPP) with kernel $L$, where the joint effects of diversity and influence are captured by $L_Y$.

## B.2 THE DETERMINANT AS A MEASURE OF SAMPLE DIVERSITY

Let $\{\mathbf{x}_1, \mathbf{x}_2, \ldots, \mathbf{x}_m\}$ be a set of $m$ samples, where each $\mathbf{x}_i \in \mathbb{R}^d$ represents a feature vector. Yhe vectors are normalized such that $\|\mathbf{x}_i\| = 1$ for all $i$. The **similarity matrix** $Y$ is defined as:

$$Y_{ij} = \mathbf{x}_i^\top \mathbf{x}_j = \langle \mathbf{x}_i, \mathbf{x}_j \rangle. \tag{15}$$

Under the normalization assumption, $Y_{ii} = 1$ for all $i$, and $Y_{ij} = \cos \theta_{ij}$, where $\theta_{ij}$ is the angle between $\mathbf{x}_i$ and $\mathbf{x}_j$.

The key insight connecting diversity to the determinant arises from the geometric interpretation of the similarity matrix. The $m$-dimensional volume of the parallelotope spanned by the vectors $\mathbf{x}_1, \ldots, \mathbf{x}_m$ is given by:

$$\text{Vol}_m(\mathbf{x}_1, \ldots, \mathbf{x}_m) = \sqrt{\det(Y)}. \tag{16}$$

Equivalently,

$$\det(Y) = \left( \text{Vol}_m(\mathbf{x}_1, \ldots, \mathbf{x}_m) \right)^2. \tag{17}$$

This can be derived by letting $X$ be the $d \times m$ matrix whose columns are $\mathbf{x}_1, \ldots, \mathbf{x}_m$. The volume is inherently $\sqrt{\det(X^\top X)}$.

We now state and prove the main theorem connecting the determinant to diversity. Let $S_{k-1} = \text{span}(\mathbf{x}_1, \ldots, \mathbf{x}_{k-1})$ denote the subspace spanned by the first $k-1$ vectors, and let $\phi_k$ be the angle between $\mathbf{x}_k$ and its orthogonal projection onto $S_{k-1}$. The volume of the parallelotope can be expressed recursively as:

$$\text{Vol}_m(\mathbf{x}_1, \ldots, \mathbf{x}_m) = \prod_{k=1}^{m} \|\mathbf{x}_k\| \cdot \prod_{k=2}^{m} \sin \phi_k = \prod_{k=2}^{m} \sin \phi_k, \tag{18}$$

where the second equality follows from the normalization $\|\mathbf{x}_k\| = 1$. Each factor $\sin \phi_k \in [0, 1]$ quantifies the orthogonal component of $\mathbf{x}_k$ relative to the previous vectors:

- $\sin \phi_k = 1$ if and only if $\mathbf{x}_k$ is orthogonal to $S_{k-1}$ (maximal orthogonal component)
- $\sin \phi_k = 0$ if and only if $\mathbf{x}_k \in S_{k-1}$ (linear dependence, zero orthogonal component)

Since $\det(Y) = \mathrm{Vol}^2 = \prod_{k=2}^m \sin^2 \phi_k$, we observe:

- $\det(Y)$ approaches its maximum value when all $\sin \phi_k \to 1$ (i.e., when each vector is orthogonal to the span of previous vectors, yielding pairwise orthogonality).
- $\det(Y)$ approaches zero when any $\sin \phi_k \to 0$, indicating linear dependence and minimal diversity.

Thus, $\det(Y)$ strictly increases as the vectors become more orthogonal.

Since $Y$ is a real symmetric matrix and represents inner products of normalized vectors, it is positive semidefinite. The diagonal entries are all $1$. By the **Hadamard inequality** for positive semidefinite matrices, the determinant is bounded by the product of the diagonal entries:

$$\det(Y) \leq \prod_{i=1}^m Y_{ii} = 1. \tag{19}$$

This establishes the upper bound. Now, we prove the equality condition. Hadamard's inequality holds with equality *if and only if* $Y$ is a diagonal matrix. A diagonal $Y$ with $Y_{ii} = 1$ means:

$$Y = I_m, \quad \text{where } I_m \text{ is the identity matrix.} \tag{20}$$

This implies $Y_{ij} = 0$ for all $i \neq j$. Since $Y_{ij} = \langle \mathbf{x}_i, \mathbf{x}_j \rangle$, this means $\mathbf{x}_i \perp \mathbf{x}_j$ for all $i \neq j$. The vectors are mutually orthogonal. From a geometric perspective, when the vectors are orthogonal, the parallelotope they span forms an $m$-dimensional hyperrectangle and achieves its maximum volume.

## C  DETAILED EXPERIMENTAL SETUP

**Models.** We run our experiments on DeepSeek-R1-Distill-Qwen-7B (DeepSeek-AI et al., 2025), DeepSeek-R1-Distill-Llama-8B (DeepSeek-AI et al., 2025), and Qwen2.5-Math-7B (Yang et al., 2024). For DeepSeek-R1-Distill-Qwen-7B and Deepseek-R1-Distill-Llama-8B, we set the context length to 16384. For Qwen2.5-Math-7B models, we set the context length to 4096, as it is the maximum context length for this model.

**Training.** Our method is implemented based on the Verl (Sheng et al., 2025) pipeline and uses vLLM (Kwon et al., 2023) for rollout. We train DeepSeek-R1-Distill-Qwen-7B and DeepSeek-R1-Distill-Llama-8B on 64×H200 GPUs, and Qwen2.5-Math-7B on 32×H200 GPUs. For training datasets, we use the DAPO-Math Yu et al. (2025) as the training dataset. During rollouts, we set the temperature to 1 and sample 8 responses per prompt. The training batch size is set to 256. We apply the on-policy GRPO algorithm to train the model. Similar to Yue et al. (2025), we remove both the KL divergence loss and the entropy loss. We train all models for 1000 steps, and we optimize the actor model using the AdamW (Loshchilov & Hutter, 2019) optimizer with a constant learning rate of 2e-6 for DeepSeek-R1-Distill-Qwen-7B and Deepseek-R1-Distill-Llama-8B and 1e-6 for Qwen-Math-7B. The actor module is optimized using Fully Sharded Data Parallel (FSDP) (Zhao et al., 2023b) for efficient distributed training. The chat template we use is "User: \n [question] \n Please reason step by step, and put your final answer within \boxed{}. \n \n Assistant:". For offline data selection, we first apply PageRank-weighted Determinantal Point Process (DPP) to reduce the sample set to 50% of its original size. We then perform difficulty-aware sampling based on a normal distribution to select the final subset, which constitutes 20% of the full dataset. The mean $\mu$ and standard deviation $\sigma$ of the difficulty distribution for the final selected subset are set to 0.5 and 0.2, respectively. In the online data selection phase, we set the window size for recent epochs $s$ to 5, the replay sample ratio $\rho$ to 0.05, and the threshold $\lambda$ for filtering out poor negative rollouts to 1.5. For the linear decay of rollout pruning, we initialize the sampling rate $\alpha_0$ to 1 and apply a decay rate $d$ of 0.05, gradually reducing the proportion of samples used for rollout and policy update until it reaches 20% of the batch size.

**Evaluation.** For evaluation benchmarks, we use three widely used complex mathematical reasoning benchmarks (*i.e.,* AIME24, AIME25, Math500 (Hendrycks et al., 2021)) and two other reasoning

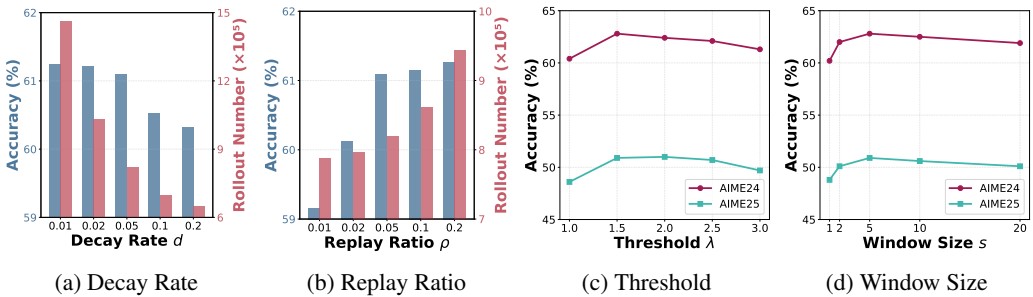

Figure 8: Hyperparameter analysis

benchmarks (*i.e.,* GPQA (Rein et al., 2023) and LiveCodeBench (Jain et al., 2025)) to evaluate the model performance. We follow Zheng et al. (2025b); Yan et al. (2026) to evaluate models on those benchmarks every 50 steps and report the performance of the checkpoint that obtains the best average performance on five benchmarks. All evaluations are conducted in a zero-shot setting. Following DeepSeek-AI et al. (2025); Wang et al. (2025c); Li et al. (2025d), we evaluate all models setting temperature to 0.6 and top-k to 0.95. We repeat the test set 32 times for evaluation stability for all benchmarks and report the average accuracy.

# D ADDITIONAL EXPERIMENTS

## D.1 THE EFFECT OF DIFFERENT SCALE MODELS

To evaluate the scaling ability of our proposed method, we conduct experiments on Qwen3-8B, Qwen3-14B, and Qwen3-32B using the DAPO algorithm. The training was conducted with a batch size of 512 and a mini-batch size of 32, resulting in 16 gradient steps per training batch. We report average performance across three mathematical reasoning benchmarks (*i.e.,* AIME24, AIME25, and Math500). As illustrated in Figure 7, we observe that our approach demonstrates strong scalability across different model scales. This improvement can be attributed to the offline data selection strategy of

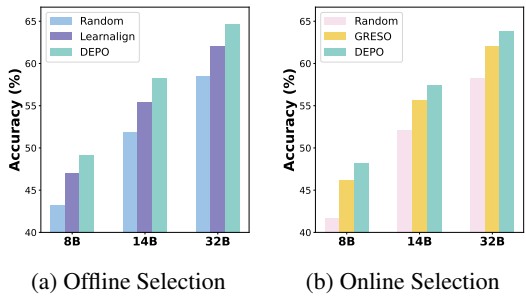

Figure 7: Different scales of LLMs.

DEPO, which selects a high-quality subset based on diversity, influence, and difficulty, which is beneficial for RLVR training. Furthermore, we select high explorability samples for rollouts and policy updates, and incorporate under-explored samples for replay, which significantly improves training efficiency without sacrificing performance.

## D.2 HYPERPARAMETER ANALYSIS

In this work, we adopt a linear decay strategy to gradually decrease the proportion of rollouts in each batch over training epochs. As shown in Figure 8a, setting the decay rate $d$ to 0.05 leads to a slight performance drop while substantially reducing rollout numbers. If the decay rate is too high, many samples may not be sufficiently trained, leading to suboptimal final performance. Conversely, an excessively low decay rate increases the number of rollouts, thereby reducing training efficiency. With respect to the replay ratio, setting it to 0.05 allows the model to achieve an optimal balance between final performance and training efficiency, as illustrated in Figure 8b. An excessively high replay ratio introduces unnecessary computational overhead due to redundant rollouts, while a ratio that is too low may prevent challenging samples from being adequately trained, thereby limiting the model's reasoning capability.

We further analyze two hyperparameters: the threshold $\lambda$ for selecting high-quality negative rollouts and the window size $s$. As shown in Figure 8c, performance initially improves and then declines

as the threshold increases. This indicates that when the threshold is set too low, potentially useful samples that could help exploration may be excluded. Conversely, an excessively high threshold may lead to the inclusion of noisy rollouts (*e.g.,* nonsensical text rollouts), which can adversely affect model performance. Regarding the window size $s$, Figure 8d indicates that the model performs best when $s = 5$. This suggests that the window size should be chosen within an appropriate range. One possible explanation is that a very small window may not capture broader historical training dynamics, while an overly large window may not focus on recent training trends.

### D.3 The Effect of Dynamic Offline Data Curation Strategy

To further investigate the effectiveness of dynamic offline curation strategies, we conduct additional experiments beyond our static difficulty-based selection with approaches that periodically update the training subset. Specifically, we implement two dynamic strategies: (1) Dividing training into two (*i.e.,* 10% data per phase, resample every 1250 minutes) and four phases (*i.e.,* 5% data per phase, resample every 625 minutes). (2) First training on a static 20% subset until convergence, then continuing with a newly resampled 10% subset.

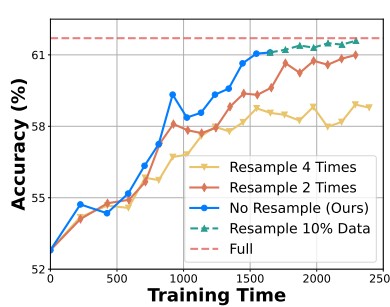

Figure 9: Dynamic Offline Curation.

Figure 9 compares the performance of these dynamic strategies against our static baseline. As we can see, updating the subset with only 5% data per phase results in performance degradation, as the model tends to overfit the current phase's data distribution and forget previously learned data. Besides, the two-phase approach strategy with 10% data per phase performs comparably to our static 20% baseline. This reveals a critical trade-off in dynamic curation between maintaining alignment with a difficulty-matched data distribution and preventing catastrophic forgetting.

Furthermore, when extending training with a new 10% subset after initial convergence on 20% data, we observe only slight improvements, indicating that data freshness through resampling is effective. However, the limited improvement suggests that the DAPO dataset is nearly saturated for this model's performance potential.

## E Detailed Description of Baselines

In this part, we provide detailed descriptions of all the baselines used in our experiments. For offline data selection methods, we compare our method with random selection, conventional supervised fine-tuning (SFT) data selection methods (*i.e.,* PPL-Top (Laurençon et al., 2022) and PPL-Middle (Ankner et al., 2025)), RLVR selection method (*i.e.,* LIMR (Li et al., 2025b) and Learnalign (Li et al., 2025a)). Table 4 presents a comparison of our method with others.

Table 4: Comparison of RLVR data selection methods.

| Method | Offline | Offline Method | Online | Online Method |
|---|---|---|---|---|
| LIMR | ✓ | Reward trends | × | – |
| One-shot RLVR | ✓ | Reward variance | × | – |
| Learnalign | ✓ | Gradient alignment | × | – |
| Polaris | ✓ | Difficulty-based | × | – |
| GRESO | × | – | ✓ | History reward variance |
| **DEPO (Ours)** | ✓ | Multi-dimension | ✓ | Explorability-guided |

- **Random**: Randomly samples data from the training set.

- **PPL-Top** (Laurençon et al., 2022): Selects the data with the highest perplexity.

- **PPL-Middle** (Ankner et al., 2025): Selects the data with the middle perplexity.

- **LIMR** (Li et al., 2025b): Selects the data whose learning patterns complement the model's overall reward trajectory.

- **Learnalign** (Li et al., 2025a): Selects the data based on representativeness (measured via gradients during warmup training) and difficulty (determined by rollout accuracy).

For online data selection methods, we incorporate them into our offline selected subset and compare against random online selection and GRESO (Zheng et al., 2025b).

- **Random**: Randomly filters 40% of the data at each batch prior to rollout during training.

- **GRESO** (Zheng et al., 2025b): Probabilistically filter historical samples with zero variance at each batch before rollout during training.

## F  RELATED WORK

### F.1  REINFORCEMENT LEARNING WITH VERIFIABLE REWARD

Reinforcement Learning with Verifiable Reward (RLVR) has emerged as a promising paradigm for enhancing the complex reasoning capabilities of large language models (LLMs), particularly in domains such as mathematics and code generation. The key advantage of this approach is its reward design, which relies solely on simple verification functions to provide binary rewards without requiring learned reward models. DeepSeek-R1 (DeepSeek-AI et al., 2025) introduces the GRPO algorithm under the RLVR framework and demonstrates its effectiveness in significantly scaling the reasoning abilities of LLMs. Building on GRPO, subsequent work have further advanced RLVR by refining various aspects, including loss functions (Liu et al., 2025a; Yu et al., 2025; Zheng et al., 2025a; Chen et al., 2025a), token-level entropy (Wang et al., 2025a; Hao et al., 2025), advantage estimation (Cheng et al., 2025; Wang et al., 2025d; Li et al., 2025c), and hyperparameter (Liu et al., 2025b; An et al., 2025; Xi et al., 2025; Liu et al., 2026). In this work, we focus on improving the data efficiency of RLVR to reduce computational costs while maintaining model performance.

### F.2  DATA EFFICIENCY FOR RLVR

Data efficiency aims to enhance model performance by strategically selecting high-quality training samples. Existing RLVR data selection approaches can be broadly categorized into offline and online strategies. Offline data selection methods focus on identifying a high-quality subset of data prior to training. Some studies select samples based on model reward trends (Li et al., 2025b), reward variance (Wang et al., 2025b), and gradient alignment (Li et al., 2025a). While effective, these methods require training the original or warmup dataset for several epochs for selection. Another line of work (An et al., 2025) uses offline rollouts to estimate sample difficulty relative to the current model's capabilities. However, this approach requires computationally expensive rollouts across the entire dataset and fails to account for the diversity and influence of the training subsets. On the other hand, online data selection methods aim to reduce training overhead by dynamically filtering samples during the training process. These approaches target samples that contribute little to improving reasoning capabilities yet require costly rollouts. GRESO (Zheng et al., 2025b) employs a probabilistic filtering strategy to exclude samples with historical zero variance. Although it improves efficiency, this approach only focuses on historical zero-variance samples and does not distinguish among other samples. In this work, we enhance the data efficiency of RLVR by integrating optimized strategies for both offline and online data selection.

## G  CASE STUDY

In this section, we provide detailed case studies on five reasoning benchmarks.

---

**A case study from AIME24**

**[Question:]**

Every morning Aya goes for a $9$-kilometer-long walk and stops at a coffee shop afterwards. When she walks at a constant speed of $s$ kilometers per hour, the walk takes her 4 hours, including $t$ minutes spent in the coffee shop. When she walks $s+2$ kilometers per hour, the walk takes her 2 hours and 24 minutes, including $t$ minutes spent in the coffee shop. Suppose Aya walks at $s+\frac{1}{2}$ kilometers per hour. Find the number of minutes the walk takes her, including the $t$ minutes spent in the coffee shop.\nPlease reason step by step, and put your final answer within \\boxed{{}}.

**[Answer:]**

Okay, so I have this problem about Aya's morning walk and her coffee stops. Let me try to figure this out step by step.\n\nFirst, let's parse the information given:\n\n1. Every morning, Aya goes for a 9-kilometer-long walk and then stops at a coffee shop.\n2. When she walks at a constant speed of 's' kilometers per hour, the entire walk, including her coffee stop, takes her 4 hours. Wait, hold on, the problem says it takes her 4 hours, but 4 hours is 240 minutes. But then it mentions including t minutes spent in the coffee shop. Hmm, maybe I need to clarify.\n\nWait, maybe the total time includes both walking and coffee time. So, when she walks at speed s km/h, her total time is 4 hours, which includes t minutes in the coffee shop.\n\nSimilarly, when she walks at s + 2 km/h, the total time is 2 hours and 24 minutes, which also includes t minutes coffee time.\n\nOur goal is to find the time it takes her when she walks at s + 1/2 km/h, again including the t minutes coffee stop.

...

**Final Answer**\n\\boxed{204}\n</think>\n\nGiven that Aya goes for a 9-kilometer-long walk every morning and stops at a coffee shop, we need to determine the total time she takes when she walks at a speed of \\( s + \\frac{1}{2} \\) km/h, including the time spent in the coffee shop.\n\nFirst, we set up the equations based on the given information:\n\n1. When walking at speed \\( s \\) km/h:\n   - Total time = 4 hours = 240 minutes.\n   - Let \\( t \\) be the time spent in the coffee shop.\n   - Equation: \\( \\frac{540}{s} + t = 240 \\).\n\n2. When walking at speed \\( s + 2 \\) km/h:\n   - Total time = 2 hours 24 minutes = 144 minutes.\n   - Equation: \\( \\frac{540}{s + 2} + t = 144 \\).\n\nSubtracting the second equation from the first to eliminate \\( t \\):\n\\[ \\frac{540}{s} - \\frac{540}{s + 2} = 96 \\]\n\n\\[ 540 \\left( \\frac{1}{s} - \\frac{1}{s + 2} \\right) = 96 \\]\n\n\\[ 540 \\left( \\frac{2}{s(s + 2)} \\right) = 96 \\]\n\n\\[ \\frac{1080}{s(s + 2)} = 96 \\]\n\n\\[ \\frac{1080}{96} = s(s + 2) \\]\n\n\\[ 11.25 = s^2 + 2s \\]\n\n\\[ s^2 + 2s - 11.25 = 0 \\]\n\nSolving the quadratic equation:\n- Discriminant: \\( 4 + 45 = 49 \\)\n- Solutions: \\( s = \\frac{-2 \\pm 7}{2} \\)\n- Positive solution: \\( s = 2.5 \\) km/h.\n\nUsing \\( s = 2.5 \\) to find \\( t \\):\n- From \\( \\frac{540}{2.5} + t = 240 \\)\n   - \\( 216 + t = 240 \\)\n   - \\( t = 24 \\) minutes.\n\nWhen walking at \\( s + \\frac{1}{2} = 3 \\) km/h:\n- Time to walk 9 km: \\( \\frac{9}{3} = 3 \\) hours = 180 minutes.\n- Total time including coffee: \\( 180 + 24 = 204 \\) minutes.\n\nThus, the total time taken is \\(\\boxed{204}\\) minutes.

---

**A case study from AIME25**

**[Question:]**

Find the sum of all integer bases $b>9$ for which $17_b$ is a divisor of $97_b.$\nPlease reason step by step, and put your final answer within \\boxed{{}}.

**[Answer:]**

Alright, let's see. I have this problem here: I need to find the sum of all integer bases \\( b > 9 \\) for which \\( 17_b \\) is a divisor of \\( 97_b \\). Hmm, okay. So, bases greater than 9, which means we're looking at bases starting from 10 upwards, right? \n\nFirst, I remember that when numbers are written in a certain base \\( b \\), their value in decimal (base 10) can be calculated. For example, a number like \\( 17_b \\) is equal to \\( 1 \\times b + 7 \\times 1 = b + 7 \\) in base 10. Similarly, \\( 97_b \\) would be \\( 9 \\times b + 7 \\times 1 = 9b + 7 \\) in base 10. \n\nSo, the problem is asking for all bases \\( b > 9 \\) where \\( 17_b \\) divides \\( 97_b \\). That means that when we convert both numbers to base 10, \\( 97_b \\) should be divisible by \\( 17_b \\) in base 10 terms.

...

**Final Answer**\n\\boxed{70}\n</think>\n\nTo solve the problem, we need to find all integer bases \\( b > 9 \\) for which \\( 17_b \\) is a divisor of \\( 97_b \\). \n\nFirst, we convert the numbers from base \\( b \\) to base 10:\n- \\( 17_b = 1 \\times b + 7 = b + 7 \\)\n- \\( 97_b = 9 \\times b + 7 = 9b + 7 \\)\n\nWe need \\( b + 7 \\) to divide \\( 9b + 7 \\). This translates to the congruence:\n\\[ 9b + 7 \\equiv 0 \\pmod{b + 7} \\]\n\nRewriting this, we let \\( n = b + 7 \\), so \\( b = n - 7 \\). Substituting back, we get:\n\n\\[ 9(n - 7) + 7 \\equiv 0 \\pmod{n} \\]\n\n\\[ 9n - 63 + 7 \\equiv 0 \\pmod{n} \\]\n\n\\[ 9n - 56 \\equiv 0 \\pmod{n} \\]\n\n\\[ -56 \\equiv 0 \\pmod{n} \\]\n\nThis implies \\( n \\) is a divisor of 56. Since \\( b > 9 \\), \\( n = b + 7 > 16 \\). The divisors of 56 greater than 16 are 28 and 56.\n\nThus, solving for \\( b \\):\n- If \\( n = 28 \\), then \\( b = 28 - 7 = 21 \\)\n- If \\( n = 56 \\), then \\( b = 56 - 7 = 49 \\)\n\nBoth 21 and 49 are valid bases as they satisfy the digit constraints in their respective numbers. Therefore, the sum of these bases is:\n\n\\[ 21 + 49 = 70 \\]\n\n\\[\n\\boxed{70}\n\\]

---

### A case study from MATH500

**[Question:]**

Define\n\\[p = \\sum_{k = 1}^\\infty \\frac{1}{k^2} \\quad \\text{and} \\quad q = \\sum_{k = 1}^\\infty \\frac{1}{k^3}.\\] Find a way to write\n\\[\\sum_{j = 1}^\\infty \\sum_{k = 1}^\\infty \\frac{1}{(j + k)^3}\\] in terms of $p$ and $q.$\nPlease reason step by step, and put your final answer within \\boxed{}.

**[Answer:]**

Okay, so I have this problem here where I need to express the double sum \\(\\sum_{j = 1}^\\infty \\sum_{k = 1}^\\infty \\frac{1}{(j + k)^3}\\) in terms of \\(p\\) and \\(q\\), where \\(p = \\sum_{k = 1}^\\infty \\frac{1}{k^2}\\) and \\(q = \\sum_{k = 1}^\\infty \\frac{1}{k^3}\\). Hmm, interesting. I remember that \\(p\\) is related to the Basel problem and equals \\(\\frac{\\pi^2}{6}\\), and \\(q\\) is Apery's constant, approximately 1.202... But maybe I don't need their exact values, just to express the double sum in terms of \\(p\\) and \\(q\\).\n\nSo, the double sum is over two indices, \\(j\\) and \\(k\\), both starting from 1 to infinity, of \\(1/(j + k)^3\\). I think I need to find a clever way to rewrite this or maybe relate it to the known series \\(p\\) and \\(q\\). Let's think about how to handle double sums. Often, double sums can sometimes be converted into single sums by clever manipulation or using integral representations.\n\nLet me write down the double sum explicitly:\n\n\\[S = \\sum_{j=1}^\\infty \\sum_{k=1}^\\infty \\frac{1}{(j + k)^3}\\]\n\nI recall that for such sums, sometimes we can change variables to simplify the expression. Maybe set \\(n = j + k\\).

...

For each \\(n\\), \\(j\\) can range from 1 to \\(n - 1\\), so we rewrite the double sum as a single sum:\n\n\\[S = \\sum_{j=1}^\\infty \\sum_{k=1}^\\infty \\frac{1}{(j + k)^3} = \\sum_{n=2}^\\infty \\sum_{j=1}^{n - 1} \\frac{1}{n^3}\\]\n\nFor each \\(n\\), there are \\(n - 1\\) pairs \\((j, k)\\), so:\n\n\\[S = \\sum_{n=2}^\\infty (n - 1) \\cdot \\frac{1}{n^3}\\]\n\nWe split the fraction:\n\n\\[\\frac{n - 1}{n^3} = \\frac{1}{n^2} - \\frac{1}{n^3}\\]\n\nThus,\n\n\\[S = \\sum_{n=2}^\\infty \\left( \\frac{1}{n^2} - \\frac{1}{n^3} \\right)\\]\n\nThis can be written as:\n\n\\[S = \\left( \\sum_{n=2}^\\infty \\frac{1}{n^2} \\right) - \\left( \\sum_{n=2}^\\infty \\frac{1}{n^3} \\right)\\]\n\nUsing the known series \\(p = \\sum_{n=1}^\\infty \\frac{1}{n^2}\\) and \\(q = \\sum_{n=1}^\\infty \\frac{1}{n^3}\\), we have:\n\n\\[\n\\sum_{n=2}^\\infty \\frac{1}{n^2} = p - 1 \\quad \\text{and} \\quad \\sum_{n=2}^\\infty \\frac{1}{n^3} = q - 1\n\\]\n\nTherefore,\n\n\\[S = (p - 1) - (q - 1) = p - q\n\\]\n\nThe final answer is: \\boxed{p - q}

## A case study from GPQA

**[Question:]**

Answer the following multiple choice question. The last line of your response should be of the following format: 'Answer: $LETTER' (without quotes) where LETTER is one of ABCD. Think step by step before answering.\n\nTwo quantum states with energies E1 and E2 have a lifetime of 10^−9 sec and 10^−8 sec, respectively. We want to clearly distinguish these two energy levels. Which one of the following options could be their energy difference so that they can be clearly resolved?\n\nA) 10^−9 eV\nB) 10^−11 eV\nC) 10^−8 eV\nD) 10^−4 eV

**[Answer:]**

Okay, so I have this quantum mechanics problem in front of me. Hmm, let's try to think it through. The question is about distinguishing two quantum states with different lifetimes. Right, I remember that in quantum mechanics, the uncertainty principle plays a role in determining the observable differences between energy levels. Or wait, perhaps it's about the Heisenberg uncertainty principle? Or maybe it's related to the line width due to the energy uncertainty affecting the transition between states.\n\nSo the problem states that two quantum states have energies E1 and E2 with lifetimes of 1e−9 seconds and 1e−8 seconds, respectively. The goal is to find which energy difference would allow us to clearly distinguish these two energy levels. The options are A to D.\n\nI'm a bit rusty on this, but I think the key here is the concept of the spectral line width. Oh wait, I remember something about the width of spectral lines being related to the uncertainty in energy, or perhaps the Heisenberg uncertainty principle. Because if the lifetimes are different, their energy differences might lead to different spectral lines that can be resolved.\n\nHeisenberg's uncertainty principle says that the product of the uncertainty in energy (\delta E) and the uncertainty in time (\delta t) is approximately h/2.\n\nAlternatively, I remember that the natural linewidth of a spectral line (due to the uncertainty principle) is related to the inverse of the lifetime. So the linewidth \deltaE is approximately h / \tau, where \tau is the lifetime. Because the uncertainty in energy \deltaE would be on the order of h/\tau.\n\nSo if we have two states with lifetimes \tau1 and \tau2, their respective spectral linewidths would be \deltaE1 = h / \tau1 and \deltaE2 = h / \tau2. So the difference in their energy levels, \deltaE = E2 − E1, needs to be larger than the sum of their linewidths, I think, to distinguish the two lines so they don't overlap and can be resolved. Because if the separation is less than the sum of their widths, they might merge into one broad peak, making them indistinguishable.\n\nWait, but I might have this reversed. Maybe the difference in energies (delta E) between the two levels should be greater than the sum of their individual energy uncertainties (line widths) to resolve them as two separate peaks. Because if the separation is larger than the sum of the two uncertainties, they can be distinguished. Otherwise, the two peaks would overlap into one broad peak.\n\nLet's clarify: For two spectral lines, each with width about \deltaE1 and \deltaE2, the energy separation needs to be greater than \deltaE1 + \deltaE2 for them to be distinguishable. Because if their separation is less than the sum of their linewidths, their peaks might overlap, making it impossible to tell them apart.\n\nSo, the energy difference \deltaE should be greater than (\deltaE1 + \deltaE2).\n\nBut wait, wait. Alternatively, perhaps the requirement is that \deltaE > (\deltaE1 + \deltaE2), but let me think about it. If two peaks are separated by more than the sum of their widths, they can be resolved. Because each line has a width (say, the full width at half maximum, FWHM), and for them to be distinct, the distance between them must be greater than the sum of their FWHMs. Otherwise, the second line would start before the first has ended, and they merge into one.

...

Alternatively, maybe the question is considering the inverse situation: the energy difference should be more than each of the individual widths, but that doesn't make sense because the larger one is 6.5 e−7, and no option meets that except D. So likely, the correct answer is D.\n</think >\n\nTo determine which energy difference allows the two quantum states to be clearly resolved, we analyze the relationship between their lifetimes and energy widths using the uncertainty principle. The natural linewidth of a spectral line is related to the lifetime of a state by the uncertainty principle: \n\n\\[\n\\Gamma = \\frac{\\hbar}{\\tau},\n\\]\n\nwhere \\( \\Gamma \\) is the linewidth and \\( \\tau \\) is the lifetime. For the two states, the lifetimes are \\( \\tau_1 = 10^{-9} \\, \\text{s} \\) and \\( \\tau_2 = 10^{-8} \\, \\text{s} \\). Their respective linewidths are:\n\n\\[\n\\Gamma_1 = \\frac{\\hbar}{\\tau_1}, \\quad \\Gamma_2 = \\frac{\\hbar}{\\tau_2}.\n\\], we calculate:\n\n\\[\n\\Gamma_1 = \\frac{6.582 \\times 10^{-16}}{10^{-9}} = 6.582 \\times 10^{-7} \\, \\text{eV},\n\\]\n\n\\[\n\\Gamma_2 = \\frac{6.582 \\times 10^{-16}}{10^{-8}} = 6.582 \\times 10^{-8} \\, \\text{eV}.\n\\]\n\nThe sum of the linewidths is:\n\n\\[\n\\Gamma_1 + \\Gamma_2 = 6.582 \\times 10^{-7} + 6.582 \\times 10^{-8} = 1.316 \\times 10^{-7} \\, \\text{eV}.\n\\]\n\nFor the two states to be clearly resolved, the energy difference \\( \\Delta E \\) between them must be greater than the sum of their linewidths:\n\n\\[\n\\Delta E > \\Gamma_1 + \\Gamma_2 \\approx 1.316 \\times 10^{-7} \\, \\text{eV}.\n\\]\n\nNow, we evaluate the given options:\n\n- **Option A: \\( 10^{-9} \\, \\text{eV} \\)** \n This is much smaller than \\( 1.316 \\times 10^{-7} \\, \\text{eV} \\), so it does not satisfy the condition.\n- **Option B: \\( 10^{-11} \\, \\text{eV} \\)** \n This is even smaller than \\( 10^{-9} \\, \\text{eV} \\), so it does not satisfy the condition.\n- **Option C: \\( 10^{-8} \\, \\text{eV} \\)** \n This is \\( 1.0 \\times 10^{-8} \\, \\text{eV} \\), which is smaller than \\( 1.316 \\times 10^{-7} \\, \\text{eV} \\), so it does not satisfy the condition.\n- **Option D: \\( 10^{-4} \\, \\text{eV} \\)** \n This is \\( 1.0 \\times 10^{-4} \\, \\text{eV} \\), which is much larger than \\( 1.316 \\times 10^{-7} \\, \\text{eV} \\), so it satisfies the condition.\n\nThus, the only energy difference that allows the two states to be clearly resolved is \\( 10^{-4} \\, \\text{eV} \\).\n\nAnswer: D

## A case study from LiveCodeBench

**[Question:]**

### Question :\nYou are given an integer sequence A=(A_1,\\ldots,A_N) of length N. Here, A_1, A_2, \\ldots, A_N are all distinct.\nWhich element in A is the second largest?\n\nInput\n\nThe input is given from Standard Input in the following format:\nN \r\nA_1 A_2 \\ldots A_{N}\n\nOutput\n\nPrint the integer X such that the X-th element in A is the second largest.\n\nConstraints\n\n\n- 2 \\leq N \\leq 100\n- 1 \\leq A_i \\leq 10^9\n- A_1, A_2, \\ldots, A_N are all distinct.\n- All input values are integers.\n\nSample Input 1\n\n4\r\n8 2 5 1\n\nSample Output 1\n\n3\r\n\nThe second largest element in A is A_3, so print 3.\n\nSample Input 2\n\n8\r\n1 2 3 4 5 10 9 11\n\nSample Output 2\n\n6\n\n### Format: Read the inputs from stdin solve the problem and write the answer to stdout (do not directly test on the sample inputs). Enclose your code within delimiters as follows.\n```python\n# YOUR CODE HERE\n```\n\n### Answer: (use the provided format with backticks)\n\n

**[Answer:]**

Okay, I need to solve this problem where I have to find the second largest element in a given integer sequence and then determine which position it occupies in the list, specifically the index in the 1-based notation. Hmm, let's think through how to approach this.\n\nWait, wait. The problem says that each element is distinct, so I don't have to worry about duplicates. The task is to find the second largest number and then output the index (but wait, the problem says the X-th element, which is the position in the array, not zero-based). Oh right, the first sample input: 8 2 5 1. The largest is 8, then 5, then 2, then 1. Wait no, wait the second sample input is 1 2 3 4 5 10 9 11. The largest is 11, then 10, and then 9. Wait, no, wait in sample input 2, the array is [1,2,3,4,5,10,9,11], the largest is 11 at position 8, the second largest is 10 at position 6. So the output is 6. So the task is to find the second largest element in the list, which is the element that is just smaller than the maximum, but it could be more than just that because there might be elements that are after the maximum which are smaller but larger than the rest.\n\nSo, the plan is :\n\n1. Read N and the list of integers.\n2. Find the maximum value in the list.\n3. Then find all elements that are less than the maximum. Among these, the second largest will be the maximum of that subset. But wait, wait, maybe the second largest is not necessarily the max of the rest. For example, in the first sample, the max is 8, the rest are 2,5,1. The max of those is 5, which is correct. In sample 2, the max is 11, the rest are 1,2,3,4,5,10,9. The max of those is 10, which is correct.\n\nSo, the approach is:

...

The approach involves finding the maximum element, then determining the second largest element from the remaining elements, and finally locating its position in the original list.\n\n### Approach\n1. **Read Input**: First, we read the input values, which include the length of the list and the list elements.\n2. **Find Maximum Value**: Compute the maximum value in the list.\n3. **Filter Candidates**: Create a sublist of all elements excluding the maximum value.\n4. **Find Second Largest**: From the filtered sublist, find the maximum value, which is the second largest element in the original list.\n5. **Determine Position**: Locate the index of this second largest element in the original list and convert it to a 1-based index for the output.\n\n### Solution Code\n```python\nn = int(input())\na = list(map(int, input().split()))\nmax_val = max(a)\ncandidates = [x for x in a if x != max_val]\nsecond_val = max(candidates)\nindex = a.index(second_val) + 1  # Convert to 1-based index\nprint(index)\n```\n\n### Explanation\n- **Reading Input**: We read the integer `n` which is the length of the list and then read the list of integers `a`.\n- **Finding Maximum**: Using Python's built-in `max()` function, we determine the largest value in the list.\n- **Filtering Candidates**: We create a new list `candidates` that excludes the maximum value to focus on the remaining elements.\n- **Finding Second Largest**: The maximum of the `candidates` list is the second largest element in the original list.\n- **Index Conversion**: The index of this second largest element is found using `index()`, which returns a 0-based index, so we add 1 to convert it to 1-based indexing before printing the result.\n\nThis approach ensures that we efficiently find the second largest element and correctly determine its position in the original list, adhering to the problem constraints and requirements.

