# OpenReview forum: "Towards High Data Efficiency in Reinforcement Learning with Verifiable Reward"
_ICLR.cc/2026/Conference — ICLR 2026 Poster_

### Official Review · Reviewer_1LYZ · 2025-10-27

**Soundness:** 3
**Presentation:** 3
**Contribution:** 3
**Rating:** 6
**Confidence:** 3

**Summary:**

This paper proposes DEPO, a two-stage data selection pipeline, to address the high training costs and low data efficiency often encountered in Reinforcement Learning with Value Response (RLVR). The process begins with offline data curation, where the data is meticulously chosen based on three distinct criteria: diversity, influence, and difficulty. Following this, the second stage, online rollout pruning, selects appropriate rollout samples by utilizing a metric focused on explorability. The empirical results demonstrate that DEPO achieves sample efficiency and faster learning performance compared to conventional methods.

**Strengths:**

* The paper proposes a comprehensive learning paradigm in RLVR that systematically integrates both an offline data curation phase and an online data collection phase.
* The metrics used for data selection during both the offline and online processes are highly refined and logically compelling.
* The efficiency and efficacy of the proposed method are validated through extensive experimental results conducted across a diverse range of datasets.

**Weaknesses:**

* Regarding the significant reduction in overall training time, it would be beneficial for the authors to clarify whether the initial cost of the offline data curation is included in this calculation. Since the graph construction and DPP application likely involve a notable overhead, quantifying this pre-computation cost would provide a more complete picture of the method's overall efficiency.
* The offline curation relies on difficulty-aware sampling from a fixed distribution determined before training. While this is shown to be effective, this dataset could become less challenging for the policy as it improves. The paper's claims on sample efficiency would be even more compelling with a brief discussion or analysis of why this static approach remains effective throughout the entire training process.

**Questions:**

* It would be very helpful if the authors could provide a quantitative measure of the specific pre-computation cost required for the offline curation stage.
* Given that the offline subset $D^{sub}$ relies on a static difficulty measurement, it would be interesting to know if the authors considered or experimented with a dynamic offline curation strategy? For instance, periodically re-sampling $D^{sub}$ to reflect the evolving capabilities of the policy might be an interesting alternative to explore.
* Could the authors provide a bit more detail on how the specific filtering threshold $\lambda=1.5$ was determined? Since this value multiplies a dynamic term that changes during training, a justification for why fixing the value of $\lambda$ is an appropriate and robust choice would be appreciated.

---

> ### Author Response · Authors · 2025-11-19
> **Rebuttal by Authors (1/2)**
>
> Thank you for your valuable contribution to reviewing our work.
> We truly appreciate your thoughtful comments and are pleased to provide our responses.
>
> In the revised PDF, all changes made in response to your comments are **highlighted in blue** for your convenience.
>
> ### **Q1: Regarding the significant reduction in overall training time, it would be beneficial for the authors to clarify whether the initial cost of the offline data curation is included in this calculation. Since the graph construction and DPP application likely involve a notable overhead, quantifying this pre-computation cost would provide a more complete picture of the method's overall efficiency.**
>
> Thank you for raising this important point regarding computational overhead.
> We agree that a comprehensive assessment of the offline curation cost is essential for evaluating the overall efficiency of our method.
>
>
> As the reviewer suggested, we have analyzed the time cost of each component in the offline data selection phase as follows:
>
> | Component  | Time(h)  |
> | -------- | -------- |
> | Graph Construction        | 0.75 ($\times$ 8.33) |
> | Pagerank-weighted DPP     | 0.09 ($\times$ 1.00) |
> | Difficulty-aware Sampling | 44.33 ($\times$ 492.56) |
>
> As we can see, the offline rollout process in difficulty-aware sampling is the **most computationally expensive** step, significantly exceeding the costs of graph construction and DPP application.
>
> To ensure transparency, we have included this table as **Table 2** in the main text and included corresponding explanations in **Section 3.4**.
>
> ### **W2: The offline curation relies on difficulty-aware sampling from a fixed distribution determined before training. While this is shown to be effective, this dataset could become less challenging for the policy as it improves. The paper's claims on sample efficiency would be even more compelling with a brief discussion or analysis of why this static approach remains effective throughout the entire training process.**
>
> Thank you for this valuable question.
> In our paper, offline data selection refers to **selecting a fixed subset from the full dataset $\mathcal{D}$**, and our experimental results show that training on this static subset **still achieves performance comparable to using the full dataset**.
>
> As you correctly pointed out, this fixed subset could indeed become less challenging as the policy improves.
> This is what we observe in our training dynamics: Figure 3 shows that around step 800, the reward approaches nearly 0.9 and plateaus on validation sets, indicating that the model has **largely saturated its learning potential** on the current data distribution.
> Further improvement would likely **require new data better aligned with its evolved capabilities**.
>
> To further address this question, we have conducted additional experiments exploring periodic resampling of $\mathcal{D}^{\text{sub}}$ during training.
> **Please see our response to Q2 for detailed results and analysis**.
>
> ### **Q1: It would be very helpful if the authors could provide a quantitative measure of the specific pre-computation cost required for the offline curation stage.**
>
> Please refer to our response to W1.

---

> ### Author Response · Authors · 2025-11-19
> **Rebuttal by Authors (2/2)**
>
> ### **Q2: Given that the offline subset $D^{sub}$ relies on a static difficulty measurement, it would be interesting to know if the authors considered or experimented with a dynamic offline curation strategy? For instance, periodically re-sampling $D^{sub}$ to reflect the evolving capabilities of the policy might be an interesting alternative to explore.**
>
> Thank you for this excellent suggestion regarding dynamic offline curation.
> We have conducted additional experiments exploring three dynamic strategies:
>
> 1. Four-phase training: Resampling 5% of data every 625 minutes.
> 2. Two-phase training: Resampling 10% of data once at 1250 minutes.
> 3. Extended training: Training first on static 20% data, then continuing with newly resampled 10% data.
>
> The results are illustrated in **Figure 9** in **Appendix F.3**.
> + Experiments 1 & 2 (Different resampling frequencies): we compare these dynamic strategies against our static 20% baseline while keeping the total data amount fixed.
> We observe that frequent updates with limited data (5% per phase) lead to noticeable forgetting, as the model **overfits to each phase's data distribution**.
> Besides, the two-phase training strategy achieved performance comparable to our static 20% baseline. This suggests that with dynamic curation, there is a **critical trade-off between maintaining alignment with data difficulty and preventing catastrophic forgetting**.
> + Experiment 3 (Extended training with fresh data): when we extend training by adding a new resampled 10% subset after the initial 20% training converged, we observed only slight performance gains.
> This result confirms that data freshness does matter, as **periodic resampling is indeed effective**.
> However, the limited improvement also indicates that the DAPO dataset is **nearly saturated** for the model's performance potential.
>
> Due to space constraints, we have added these experiments and detailed discussions to **Appendix F.3** in our revised version.
>
>
> ### **Q3: Could the authors provide a bit more detail on how the specific filtering threshold was determined? Since this value multiplies a dynamic term that changes during training, a justification for why fixing the value of is an appropriate and robust choice would be appreciated.**
>
> Since the entropy of the model changes throughout training, precisely defining an absolute "good" rollout entropy value is challenging.
> Therefore, we use the mean entropy of positive rollouts as a stable reference, as it consistently reflects the desired entropy range for high-quality rollouts.
> To account for the fact that negative rollouts generally exhibit higher entropy, we allow those with strong exploratory potential to have moderately high entropy and use a fixed $\lambda$ to bound the acceptable range.
>
> Furthermore, as shown in **Figure 8c** in **Appendix F.2**, we conducted a hyperparameter study on the negative rollout filtering threshold $\lambda$ ranging from 1.0 to 3.0.
> Performance peaks at $λ=1.5$, indicating that a threshold set too low may exclude useful exploratory samples, while higher values may introduce noisy rollouts and hurt performance.

---

> > ### Comment · Reviewer_1LYZ · 2025-11-25
> >
> > Thank you for your detailed responses.
> > However, I still have two remaining concerns that I would like to see addressed:
> > 1. Regarding efficiency:
> >
> > My concern is that if the offline data curation process is computationally expensive, potentially taking as much time as the training reported in Table 3, the proposed method might be achieving efficiency at the expense of significant pre-processing time.
> > **Could you clarify how the time cost of curation compares to the training time?**
> >
> > 2. Regarding Appendix F.3:
> >
> > I appreciate the additional experiments in Appendix F.3.
> > The results suggest that the proposed method is already efficient and that increasing the resampling phases leads to overfitting due to the reduced data per phase.
> > This leads to a follow-up question: What would be the effect of progressively reducing the dataset size at each resampling step?
> > For instance, in a 4-phase resampling setup, using a sequence like 20, 15, 10, 5 for the data per phase.
> > **Please note that I am primarily interested in your intuition regarding this hypothesis, rather than requesting a full experiment.**
> > Given the limited rebuttal duration, strictly running this experiment is not required; I would appreciate your insights on how such a schedule might affect the performance.

---

> > > ### Author Response · Authors · 2025-11-25
> > > **Further Response to Reviewer ILYZ**
> > >
> > > We sincerely thank you for the time and effort you have dedicated to reviewing our rebuttal and for your thoughtful questions, which have provided us with valuable opportunities to further improve our work.
> > >
> > > ### **Q1: My concern is that if the offline data curation process is computationally expensive, potentially taking as much time as the training reported in Table 3, the proposed method might be achieving efficiency at the expense of significant pre-processing time. Could you clarify how the time cost of curation compares to the training time?**
> > >
> > > Thank you for raising this important point.
> > > We appreciate the chance to clarify the computational cost of our offline data curation process.
> > >
> > > Due to limited resources, the offline data curation was done on **4×RTX 3090 GPUs (4 $\times$ 24G)**, while the RL training was conducted on **64×H200 GPUs (64 $\times$ 140G)**.
> > > This **hardware difference** explains why the offline curation time reported in Table 2 appears long, where the difficulty-aware sampling process takes about two days.
> > >
> > > In our experiments, the rollout process accounts for about 70% of the total RLVR training time.
> > > If we use the same GPU resources for rollout as for training, performing rollout on 50% training data  (around 70k rollout numbers) would take about **1.75 hours**, which **equals about 25 RL training steps**.
> > > Since the **full RL training requires about 1000 steps**, the offline rollout time is actually very small in comparison.
> > >
> > > To ensure clarity, we have updated the caption of **Table 2** to: **Computational time on 4×RTX 3090 GPUs** and **highlighted it in blue** in the revised version.
> > >
> > >
> > > ### **Q2: I appreciate the additional experiments in Appendix F.3. The results suggest that the proposed method is already efficient and that increasing the resampling phases leads to overfitting due to the reduced data per phase. This leads to a follow-up question: What would be the effect of progressively reducing the dataset size at each resampling step? For instance, in a 4-phase resampling setup, using a sequence like 20, 15, 10, 5 for the data per phase. Please note that I am primarily interested in your intuition regarding this hypothesis, rather than requesting a full experiment. Given the limited rebuttal duration, strictly running this experiment is not required; I would appreciate your insights on how such a schedule might affect the performance.**
> > >
> > > Thank you for your positive feedback on the additional experiments in Appendix F.3.
> > > We also appreciate your insightful question regarding progressively reducing the dataset size across resampling phases.
> > >
> > > Based on our current experimental setup (training DeepSeek-R1-Distilled-Qwen-7B on the DAPO dataset), we believe that gradually decreasing the data volume (e.g., from 20% to 15%, 10%, and 5% over four phases) **may not lead to significant improvements**.
> > > This is because using only 20% of the data already brings the model's performance **close to the upper limit achievable on this dataset**.
> > > As shown in Appendix F.3, adding an extra 10% of resampled data in this setting only yields **marginal gains**, with results approaching those of full-dataset training.
> > >
> > > However, we hypothesize that **in an ideal scenario with a much larger and more diverse dataset**, a well-designed active data curriculum could help improve performance.
> > > If each phase selects data based on diversity, influence, and appropriate difficulty, the RL process at each stage could effectively **increase the reward** on those samples.
> > > Besides, since we use an on-policy RL algorithm with **strong generalization ability**, reward improvement on the current data often indicates that the model can solve more unseen problems of a similar type.
> > > As the model's overall capability improves, the next phase would naturally require more challenging samples, which could **iteratively** improve the model's reasoning ability.
> > >
> > > Furthermore, we believe the key factor is not **necessarily a decreasing data schedule**, but whether **each phase contains a sufficient number of samples that match the model's current learning needs**.
> > > This is similar to our findings in Appendix F.3: using only 5% of data per phase (about 900 samples) leads to a clear performance drop due to insufficient data, while increasing it to 10% (about 1,800 samples) effectively mitigates this issue.
> > >
> > >
> > > We also note that as data selection shifts toward harder examples, **longer context windows** may be needed to maintain performance.
> > > In addition, collecting **large high-quality** datasets also remains a practical challenge.
> > > Due to the limited rebuttal period, we have not been able to conduct full experiments to validate these ideas, but we hope our analysis provides useful insight.
> > >
> > >
> > > Thank you again for your engaging questions.
> > > We would be happy to discuss further if you have any additional questions.
> > >
> > > Best Regards,
> > >
> > > Authors

---

### Official Review · Reviewer_Gqob · 2025-10-31

**Soundness:** 3
**Presentation:** 3
**Contribution:** 2
**Rating:** 4
**Confidence:** 2

**Summary:**

This paper addresses the high data and computational costs associated with Reinforcement Learning with Verifiable Rewards (RLVR) for enhancing the reasoning capabilities of Large Language Models (LLMs). The authors propose DEPO (Data-Efficient Policy Optimization), a pipeline that combines offline data curation and online rollout pruning. In the offline stage, a PageRank-weighted Determinantal Point Process is used to select a diverse and influential data subset, which is then refined to favor samples of moderate difficulty aligned with the model's current capabilities. In the online stage, an "explorability" metric is used to dynamically skip rollouts for low-potential samples, thereby improving computational efficiency.

**Strengths:**

1, The proposed method, DEPO, achieves performance nearly equivalent to GRPO training on the full dataset while using only 20% of the data and significantly reducing training time. Specifically, on the DeepSeek-R1-Distill-Qwen-7B model, the full DEPO pipeline achieved a comparable average accuracy (61.1% vs. 61.7%) using only 57% of the training time and 40% of the rollouts compared to training on the full dataset (Table 1). This is a substantial practical contribution that could lower the barrier to entry for RLVR training.

2, The paper presents one of the first approaches to systematically integrate both offline data selection and online rollout pruning. This dual optimization strategy, which considers diversity, influence, and difficulty offline (Section 3.2) while performing dynamic sample filtering online (Section 3.3), is a distinguishing feature compared to prior work that has typically focused on only one of these aspects.

3, The effectiveness of the proposed method is validated across three different LLMs and five reasoning benchmarks (Table 1). The paper provides extensive empirical evidence to support its claims, including sensitivity analyses on varying dataset sizes (Figure 4), sampling ratios (Figure 5a), and rollout ratios (Figure 5b), as well as a clear ablation study demonstrating the contribution of each component (Table 2).

**Weaknesses:**

1, The core components of the proposed method are largely a combination of well-established ideas. The use of Determinantal Point Processes (DPP) and PageRank for offline selection (Section 3.2.1), and difficulty-based sampling aligned with model capability (i.e., curriculum learning) (Section 3.2.2) are not new concepts. The entropy-based "explorability" metric for online pruning (Equation 4) appears to be an intuitive extension of existing RL techniques that leverage entropy to encourage exploration. While the authors claim the "integration" of offline and online strategies as a key contribution, the lack of originality in the individual components limits the overall novelty of the work.

2, The most significant weakness is the lack of statistical validation for the experimental results. Most results, including the main ones in Table 1, present figures from a single run without reporting means and standard deviations (or confidence intervals) over multiple seeds. For instance, the performance difference between DEPO-Offline and the Full dataset on the Deepseek-R1-Distill-Qwen-7B model is a mere 0.3 percentage points (61.4 vs. 61.7). It is impossible to determine if this difference is statistically significant or simply due to random variance. This severely undermines the credibility of the paper's central claim of "maintaining comparable performance while improving efficiency."

3,The "explorability" metric (Equation 4) is intuitive but designed in a somewhat heuristic manner. The paper lacks a theoretical justification or deeper analysis for the specific combination of normalized reward (Ât), entropy, and a hard threshold (λ).

**Questions:**

1, To assess the reliability of the core results, could you please repeat the key experiments in Table 1 (e.g., for the DeepSeek-R1-Distill-Qwen-7B model) using at least three different random seeds and report the mean and standard deviation? This is crucial to verify whether the performance difference between your method (DEPO) and full-dataset training is statistically insignificant.

2, The offline "Difficulty-aware Normal Distribution Sampling" strategy (Section 3.2.2) statically prioritizes moderately difficult samples. This overlooks a comparison with a simpler, dynamic curriculum learning strategy (e.g., moving from easy to hard samples as training progresses). Could you provide a comparative experiment against such a classical curriculum strategy to demonstrate the superiority of your proposed sampling method?

3, The "explorability" metric (Equation 4), central to your online pruning, combines normalized reward (Ât) and entropy (E(o_i^t)). To isolate the contribution of these two factors, could you add to your ablation study (Table 2) a variant that performs pruning based solely on entropy, removing the reward term (Ât)? This would help determine whether the combination with the reward signal is essential.

4, The offline data curation is a two-step process: DPP pruning to 50%, followed by difficulty sampling to 20%. What is the advantage of this two-step approach over a single-step method that could, for instance, incorporate a difficulty score directly into the DPP kernel? Could you provide an experimental comparison or a stronger justification for this design choice?

---

> ### Author Response · Authors · 2025-11-19
> **Rebuttal by Authors (1/3)**
>
> Thank you for your thorough review and valuable suggestions.
> We have carefully addressed each of your questions in our response below.
>
> In the revised PDF, all corresponding changes have been **highlighted in green** for your convenience.
>
> ### **W1: The core components of the proposed method are largely a combination of well-established ideas. The use of Determinantal Point Processes (DPP) and PageRank for offline selection (Section 3.2.1), and difficulty-based sampling aligned with model capability (i.e., curriculum learning) (Section 3.2.2) are not new concepts. The entropy-based "explorability" metric for online pruning (Equation 4) appears to be an intuitive extension of existing RL techniques that leverage entropy to encourage exploration. While the authors claim the "integration" of offline and online strategies as a key contribution, the lack of originality in the individual components limits the overall novelty of the work.**
>
> While DPP, PageRank, and curriculum learning are established concepts, our **specific design** and **its integration** for RLVR training are novel and effective.
> Below, we summarize the key aspects of our contribution:
>
> + **First Integrated Data Selection Framework for RLVR:** we are the **first to integrate offline and online** data selection strategies, which contributes to more efficient RLVR training. Existing approaches optimize either offline or online selection, leading to suboptimal efficiency.
> + **Multi-dimensional Offline Curation:** Our offline selection is more than combining existing methods. We employ a **multi-dimensional** data curation approach that accounts for diversity, influence, and difficulty. This differs from existing works that rely on a **single** metric.
> + **Fine-grained Online Selection with Dynamic Replay:** Our online explorability metric is not a simple entropy extension. It incorporates historical training dynamics over a sliding window and combines advantage-weighted entropy to prioritize **medium-difficulty**, **high-potential** samples. Additionally, we introduce a **dynamic replay mechanism** to prevent under-explored samples from being overlooked.
>
> To clearly compare our approach among existing methods, we provide the following comparative summary (now included as **Table 1 in the introduction**):
>
>
> | Method            | Offline   | Offline Method | Online  | Online Method|
> |:--------          |:--------: | :--------:     | :------:|:--------:    |
> | **LIMR** [1]          |  ☑️    | Reward trends     | ✖️ | - |
> | **One-shot RLVR** [2] |  ☑️    | Reward variance   | ✖️ | - |
> | **Learnalign** [3]    |  ☑️    | Gradient alignment| ✖️ | - |
> | **Polaris**  [4]      |  ☑️    | Difficulty-based  | ✖️ | - |
> | **GRESO**    [5]      |  ✖️    |  -                | ☑️ | History Reward |
> | **DEPO (Ours)**   |  ☑️    |  Multi-dimension  | ☑️ | Explorability-guided |
>
>
> To better highlight our innovations and contributions, we have summarized these points in the **Introduction Section**.
>
> [1] LIMR: Less is More for RL Scaling, arXiv 2025.
>
> [2] Reinforcement Learning for Reasoning in Large Language Models with One Training Example, NIPS 2025.
>
> [3] LearnAlign: Reasoning Data Selection for Reinforcement Learning in Large Language Models Based on Improved Gradient Alignment, arXiv 2025.
>
> [4] POLARIS: A POst-training recipe for scaling reinforcement Learning on Advanced ReasonIng modelS, Blog 2025.
>
> [5] Act Only When It Pays: Efficient Reinforcement Learning for LLM Reasoning via Selective Rollouts, NIPS 2025.

---

> ### Author Response · Authors · 2025-11-19
> **Rebuttal by Authors (2/3)**
>
> ### **W2: The most significant weakness is the lack of statistical validation for the experimental results. Most results, including the main ones in Table 1, present figures from a single run without reporting means and standard deviations (or confidence intervals) over multiple seeds. For instance, the performance difference between DEPO-Offline and the Full dataset on the Deepseek-R1-Distill-Qwen-7B model is a mere 0.3 percentage points (61.4 vs. 61.7). It is impossible to determine if this difference is statistically significant or simply due to random variance. This severely undermines the credibility of the paper's central claim of "maintaining comparable performance while improving efficiency."**
>
> Thank you for raising this important point regarding statistical validation. We appreciate the opportunity to clarify our evaluation protocol.
>
> All results reported in our experiments are based on **32** independent evaluations, **not single runs**.
> And we report the **average accuracy** across these evaluations.
> As described in **Appendix E**, we follow the official evaluation protocol of DeepSeek-R1 [1] by setting the temperature to 0.6 and top-k to 0.95, and repeating inference 32 times across all benchmarks to ensure stability.
>
> As the reviewer suggested, we have added **standard deviations** to all results and conducted **statistical significance tests** in **Table 1** (now is **Table 3** in the revised version).
> These statistical validations confirm that the performance differences are statistically meaningful, further demonstrating the effectiveness of our method.
>
> [1] DeepSeek-R1: Incentivizing Reasoning Capability in LLMs via Reinforcement Learning.
>
>
>
> ### **W3: The "explorability" metric (Equation 4) is intuitive but designed in a somewhat heuristic manner. The paper lacks a theoretical justification or deeper analysis for the specific combination of normalized reward (Ât), entropy, and a hard threshold (λ).**
>
> As established in prior works [1,2], entropy is a well-founded indicator of exploration potential.
> In our design, we extend this concept by combining it with a normalized reward term (i.e., defined as the sum of absolute advantages $\sum_{i=1}^G|A_i|$), which peaks when sample accuracy is near 0.5 and drops to 0 when accuracy is 0 or 1.
> This term effectively prioritizes medium-difficulty samples during online selection and helps filter out samples with zero reward variance, thereby improving training efficiency.
>
> Thus, our explorability metric captures both the **exploration potential** (via entropy) and the **sample difficulty** (via absolute advantage) of each sample.
> We have added this explanation in **Section 3.1.1**.
>
> To empirically validate the design, we conducted ablation studies removing each component.
> As shown in the table below (now included as **Table 4**), both components contribute significantly to final performance:
>
>
> | Dataset  | AIME 24  | AIME 25  | MATH 500 |
> | -------- | -------- | -------- | -------- |
> | DEPO     | **62.8**    | **50.9**     | **95.9**     |
> |w/o Explorability Measurement | 58.7 | 45.3 | 93.1 |
> |w/o Normalized reward| 61.9 | 49.5 | 95.5 |
> |w/o Entropy          | 60.6 | 48.4 | 94.6 |
>
> Regarding the hard threshold $\lambda$, we performed a hyperparameter analysis in **Appendix F.2** and found that $λ=1.5$ yields the best results.
> As illustrated in Figure 8c, performance initially improves and then declines with increasing $\lambda$.
> This suggests that the threshold set too low may exclude useful exploratory samples, while values too high may introduce noisy or uninformative rollouts, ultimately harming model performance.
>
> [1] Beyond the 80/20 Rule: High-Entropy Minority Tokens Drive Effective Reinforcement Learning for LLM Reasoning, NIPS 2025.
>
> [2] The Surprising Effectiveness of Negative Reinforcement in LLM Reasoning, NIPS 2025.
>
> ### **Q1: To assess the reliability of the core results, could you please repeat the key experiments in Table 1 (e.g., for the DeepSeek-R1-Distill-Qwen-7B model) using at least three different random seeds and report the mean and standard deviation? This is crucial to verify whether the performance difference between your method (DEPO) and full-dataset training is statistically insignificant.**
>
> We would like to clarify that, as stated in **Appendix E** of our paper: **"We repeat the test set 32 times for evaluation stability for all benchmarks and report the average accuracy."**.
> This means that each test sample was evaluated 32 times, and we reported the mean accuracy across these repetitions.
>
> As the reviewer suggested, we have now updated **Table 1** to include standard deviations and performed statistical significance tests.
> These additions further confirm the reliability and statistical validity of our results.

---

> ### Author Response · Authors · 2025-11-19
> **Rebuttal by Authors (3/3)**
>
> ### **Q2: The offline "Difficulty-aware Normal Distribution Sampling" strategy (Section 3.2.2) statically prioritizes moderately difficult samples. This overlooks a comparison with a simpler, dynamic curriculum learning strategy (e.g., moving from easy to hard samples as training progresses). Could you provide a comparative experiment against such a classical curriculum strategy to demonstrate the superiority of your proposed sampling method?**
>
> As shown in our original **Figure 6**, we had already compared our method against random and stratified sampling.
> Random sampling performed worst, followed by stratified sampling, while our method achieved the best results.
> This aligns with our observation that extremely easy or hard samples provide limited training signals for RLVR.
>
> As the reviewer suggested, we have now added a comparison with a standard easy-to-hard curriculum learning strategy in **Figure 6(a)**.
> The results show that while the curriculum approach outperforms random sampling, it is still less effective than stratified sampling and significantly underperforms our method.
> From **Figure 6(a) in the updated version**, we observe that the curriculum strategy leads to **slow initial progress**, as overly simple samples provide limited learning signals.
> Performance **improves more noticeably** in the mid-phase when moderately difficult samples are introduced.
> However, the method **fails to achieve high final accuracy**, as overly difficult samples also contribute little to learning.
>
> We have added the accuracy curve and corresponding analysis to **Figure 6(a)** and **Section 4.3** to further demonstrate the effectiveness of our proposed approach.
>
> ### **Q3: The "explorability" metric (Equation 4), central to your online pruning, combines normalized reward (Ât) and entropy (E(o_i^t)). To isolate the contribution of these two factors, could you add to your ablation study (Table 2) a variant that performs pruning based solely on entropy, removing the reward term (Ât)? This would help determine whether the combination with the reward signal is essential.**
>
> Please refer to our response to W3.
> As the reviewer suggested, we conducted the ablation study below and added to **Table 4** in the paper.
>
> | Dataset  | AIME 24  | AIME 25  | MATH 500 |
> | -------- | -------- | -------- | -------- |
> | DEPO     | **62.8**     | **50.9**     | **95.9**     |
> |w/o Explorability Measurement | 58.7 | 45.3 | 93.1 |
> |w/o normalized reward| 61.9 | 49.5 | 95.5 |
> |w/o entropy          | 60.6 | 48.4 | 94.6 |
>
> ### **Q4: The offline data curation is a two-step process: DPP pruning to 50%, followed by difficulty sampling to 20%. What is the advantage of this two-step approach over a single-step method that could, for instance, incorporate a difficulty score directly into the DPP kernel? Could you provide an experimental comparison or a stronger justification for this design choice?**
>
> Thank you for this great question regarding our two-step offline selection design. We are pleased to provide both empirical and computational analysis for this approach.
> Our two-step offline data selection strategy offers significant advantages in both effectiveness and efficiency.
>
> + **Effectiveness**: In RLVR data selection, difficulty is a crucial factor. As shown in our ablation studies (**Table 4**), removing difficulty-aware sampling leads to a **substantial performance drop** compared to other components. Therefore, applying it in the second stage ensures it strongly influences the final selected subset.
>
> + **Efficiency**: Difficulty-aware sampling requires offline rollouts, which are **computationally expensive** compared to DPP pruning. By first reducing the dataset to 50% via DPP, we significantly improve overall curation efficiency. The computational costs below (measured on 4×24G GPUs) illustrate this:
>
> | Method   | Time(h)  |
> | -------- | -------- |
> | Graph Construction        | 0.75 (✖️ 8.33) |
> | Pagerank-weighted DPP     | 0.09 (✖️ 1.00) |
> | Difficulty-aware Sampling | 44.33 (✖️ 492.56) |
>
> As the reviewer suggested, we evaluate a single-step alternative that integrates difficulty directly into the DPP kernel.
> The results below confirm that this single-step approach still underperforms compared to DEPO.
>
> | Dataset  | AIME 24  | AIME 25  | MATH 500 |
> | -------- | -------- | -------- | -------- |
> | DEPO     | **62.8**     | **50.9**     | **95.9**     |
> |w/o Pagerank-weighted DPP     | 62.1 | 50.0 | 95.6 |
> |w/o Difficulty-aware Sampling | 60.3 | 47.8 | 95.1 |
> |Difficulty and Pagerank-weighted DPP          | 61.6 | 50.2 | 95.4 |

---

> ### Author Response · Authors · 2025-11-27
> **Kindly request your feedback before the end of the discussion period**
>
> Dear Reviewer Gqob,
>
> Thank you very much for taking the time to review our paper and for providing valuable suggestions.
> As the rebuttal period is coming to an end, we would like to kindly check whether our responses have adequately addressed your concerns.
>
> In response to your questions, we have made the following revisions:
> + We have **``clarified the contributions of our work``** in the **Introduction section** of the revised version.
> + We have added **``standard deviations and statistical significance``** tests in **Table 3**.
> + We have included **``additional ablation study of explorability metric``** in **Table 4**.
> + We have supplemented the **``easy-to-hard curriculum``** method in **Figure 6(a)** and the **``Difficulty and Pagerank-weighted DPP``** in **our response to Q4**.
> + We have added **``computational time``** results in **Table 2**.
>
> We believe our responses have fully addressed your questions.
> If you have any further concerns, please feel free to let us know.
> We would be happy to provide additional clarification.
>
> Best regards,
>
> Authors

---

> > ### Comment · Reviewer_Gqob · 2025-11-28
> >
> > I thank the authors for their detailed response and the extensive additional experiments conducted during the rebuttal phase.
> >
> > I appreciate that my primary concern regarding the statistical validation of the results has been addressed by providing standard deviations and significance tests. This significantly strengthens the credibility of the paper's core claim regarding efficiency. Furthermore, the inclusion of the requested ablation study (isolating the reward term) and the comparison with the dynamic curriculum strategy clearly demonstrates the necessity and superiority of the proposed pipeline components. The justification for the 2-step offline curation process is also well-supported by the computational cost analysis provided.
> >
> > In light of these improvements and clarifications, I am raising my Rating to 6 and my Confidence to 4.
> >
> > (Note on Score Update: I am currently experiencing technical difficulties editing the scores on the system directly. I will update the rating and confidence scores as soon as the feature becomes available.)

---

> > > ### Author Response · Authors · 2025-11-28
> > > **Thank you for raising the score!**
> > >
> > > Dear Reviewer Gqob,
> > >
> > > Thank you once again for reviewing our rebuttal and for your positive recognition of our responses and additional experiments. We truly appreciate your encouraging feedback and are very glad that we have been able to address all the issues you raised.
> > >
> > > We are also sincerely grateful for the many constructive suggestions you provided, including adding standard deviations and significance tests, as well as conducting ablation studies on the explorability metric and the easy-to-hard curriculum. These insightful comments have **undoubtedly helped make our paper more rigorous and comprehensive**.
> > >
> > > Best regards,
> > >
> > > Authors

---

### Official Review · Reviewer_L77o · 2025-11-01

**Soundness:** 2
**Presentation:** 2
**Contribution:** 2
**Rating:** 4
**Confidence:** 3

**Summary:**

This paper addresses improving sample efficiency in Reinforcement Learning with Verifiable Reward (RLVR). It introduces DEPO, a method that integrates offline and online strategies to reduce the data required during finetuning. DEPO is evaluated on three pre-trained models across three mathematics benchmarks, compared with baselines --- both offline and online variants of RLVR, and demonstrates strong performance.

**Strengths:**

1. The experimental results are comprehensive. Many ablation studies of different components, different hyperparameters are done to clearly show the contributions and robustness of the proposed components.

**Weaknesses:**

1. The writing could be improved, particularly in the methodology section. Additional clarification questions are provided below.
2. The work introduces many components—three in the offline phase and one in the online phase—but the related work for each component is not sufficiently discussed. For example, while the introduction touches on high-level prior work for both phases, detailed, technical comparisons are missing. This makes it somewhat difficult for reviewers to assess the novelty of the proposed algorithm.

**Questions:**

1. Related works do not sufficiently cover the literature. For instance, for RLVR, the reviewer believes that after DeepSeek-R1, there are many representative works that have used it in different places. But there are only about 5 papers are cited.
2. What is $\mathcal{B}$ in equation (5)?
3. Could you give a more detailed explanation for the exploration metric in (4)? The equation is too complicated without an intuitive explanation to validate each term.
4. In equation (6), what does top-α_e% means, which term should be the top of which set? The same for "bottom-ρ%"
5. In the final objective function (5), the reviewer didn't see the role of the final offline dataset D^{sub} if the reviewer didn't miss something. So what are the roles of the offline and online phase?
6. Could the author explain more about this claim/intuition around line 177: "To promote diversity, we use Determinantal Point Process (DPP) (Kulesza & Taskar, 2012a) to identify a subset that maximizes the determinant of the corresponding similarity submatrix: maxY ⊆P det(Y ), where Y is the submatrix of the full similarity matrix P, and det(·) represents its determinant value. This refers to selecting samples that form a larger volume in the feature space, with a larger volume indicating greater diversity." Why a larger determinant of the similarity matrix P may lead to diversity? Since intuitively, if some entries in P_ij is small, that means the corresponding sample i and j are different, giving more diversity. So should we just find those regions with small elements in P? Why the determinant is related to the diversity of samples?

---

> ### Author Response · Authors · 2025-11-19
> **Rebuttal by Authors (1/3)**
>
> We sincerely thank the reviewer for the valuable time and effort dedicated to reviewing our paper.
> We appreciate your insightful comments and constructive feedback.
>
> In response to your questions, we have carefully addressed them below.
> All corresponding modifications have been **highlighted in yellow** in the updated PDF for your easy reference.
>
> ### **W1: The writing could be improved, particularly in the methodology section. Additional clarification questions are provided below.**
>
> In response to your suggestions, we have carefully revised our writing to improve clarity and address any potential ambiguities.
> We hope these revisions can adequately address your concerns and significantly improve the readability of our paper.
>
>
> ### **W2: The work introduces many components—three in the offline phase and one in the online phase—but the related work for each component is not sufficiently discussed. For example, while the introduction touches on high-level prior work for both phases, detailed, technical comparisons are missing. This makes it somewhat difficult for reviewers to assess the novelty of the proposed algorithm.**
>
> We agree that a more detailed technical comparison will help clarify the methodological novelty of our method.
> Through an extensive literature review, we have identified the following relevant works in RLVR data selection, which are mentioned in our paper:
>
> + Offline data selection: LIMR [1], One-shot RLVR [2], LearnAlign [3], and Polaris [4].
> + Online data selection: Only GRESO [5].
>
> To clearly illustrate the technical distinctions and better highlight our methodological novelty, we have added a comprehensive comparative table (see **Table 1 in the Introduction Section**) as follows:
>
> | Method            | Offline   | Offline Method | Online  | Online Method|
> |:--------          |:--------: | :--------:     | :------:|:--------:    |
> | **LIMR** [1]          |  ☑️    | Reward trends     | ✖️ | - |
> | **One-shot RLVR** [2] |  ☑️    | Reward variance   | ✖️ | - |
> | **Learnalign** [3]    |  ☑️    | Gradient alignment| ✖️ | - |
> | **Polaris**  [4]      |  ☑️    | Difficulty-based  | ✖️ | - |
> | **GRESO**    [5]      |  ✖️    |  -                | ☑️ | History reward variance |
> | **DEPO (Ours)**   |  ☑️    |  Multi-dimension  | ☑️ | Explorability-guided |
>
> Our main contributions can be summarized as:
>
> 1. **First Integrated Approach**: DEPO is the first method to integrate both offline and online data selection, enabling more efficient RLVR training.
> 2. **Multi-dimensional Offline Selection**: Unlike prior methods relying on a single criterion, we introduce a multi-dimensional offline data selection strategy considering diversity, influence, and difficulty.
> 3. **Fine-grained Online Assessment**: GRESO's limitation lies in its sole reliance on historical zero-variance samples, which limits its effectiveness when such samples are scarce. In contrast, our explorability metric provides a fine-grained, sample-level evaluation of exploration potential.
>
> We have summarized these contributions in the **Introduction Section** to better highlight the difference between our work and previous approaches.
>
> [1] LIMR: Less is More for RL Scaling, arXiv 2025.
>
> [2] Reinforcement Learning for Reasoning in Large Language Models with One Training Example, NIPS 2025.
>
> [3] LearnAlign: Reasoning Data Selection for Reinforcement Learning in Large Language Models Based on Improved Gradient Alignment, arXiv 2025.
>
> [4] POLARIS: A POst-training recipe for scaling reinforcement Learning on Advanced ReasonIng modelS, Blog 2025.
>
> [5] Act Only When It Pays: Efficient Reinforcement Learning for LLM Reasoning via Selective Rollouts, NIPS 2025.
>
>
> ### **Q1: Related works do not sufficiently cover the literature. For instance, for RLVR, the reviewer believes that after DeepSeek-R1, there are many representative works that have used it in different places. But there are only about 5 papers are cited.**
>
> Thank you for raising this important point regarding the coverage of related work. We appreciate the opportunity to clarify and improve our literature review.
>
> In our initial submission, we did in fact include a broader set of representative RLVR works in **Appendix H**.
> + For RLVR methods, we cited GRPO along with papers focusing on **component-level optimizations** (including loss functions, token-level entropy, advantage estimation, and hyperparameters), covering **6** relevant papers.
> + Regarding data efficiency in RLVR, we referenced **5** additional papers, as mentioned in our response to W2.
>
> We fully acknowledge the rapid development of the RLVR field over the past year.
> As the reviewer suggested, we have now expanded **Section 2** (**Reinforcement Learning with Verifiable Reward**) by incorporating **5** additional recent works.
> We believe these additions provide a more comprehensive overview of recent advances in RLVR.

---

> ### Author Response · Authors · 2025-11-19
> **Rebuttal by Authors (2/3)**
>
> ### **Q2: What is $\mathcal{B}$ in equation (5)?**
>
> In Equation (5), $\mathcal{B}$ denotes the **set of samples in a training batch from the offline selected subset $\mathcal{D}^{\text{sub}}$**.
> We have revised **Section 3.3.2** to include this explicit definition to avoid any confusion.
>
>
> ### **Q3: Could you give a more detailed explanation for the exploration metric in (4)? The equation is too complicated without an intuitive explanation to validate each term.**
>
> The explorability metric in Equation (4) aims to identify which training samples are **most beneficial for encouraging exploration** during RLVR training.
> Below, we provide a step-by-step breakdown of its design.
>
> + **Step 1: Quality Filtering**: A key challenge is that while high entropy generally promotes exploration, some high-entropy negative samples (e.g., pathological trajectories) can be too noisy and destabilize training.
> To address this, we use an indicator function:
> $I(q, a, o_i^t) = \mathbb{I} \left[ \mathcal{V}(o_i^t, a) = 1 \lor \left(\mathcal{V}(o_i^t,a)=0 \land e(o_i^t) \leq \lambda e(\overline{o_i^{t+}}) \right) \right]$
> Specifically, it keeps all positive rollouts (where $\mathcal{V}(o_i^t, a)=1$),  and negative rollouts only if their entropy $e(o_i^t)$ does not exceed $\lambda$ times the mean entropy of positive rollouts $e(\overline{o_i^{t+}})$.
> Here, $(q, a)$ denotes the question-answer pair of the sample, and $o_i^t$ refers to the $i$-th rollout in epoch $t$.
> The verification function $\mathcal{V}$ returns 1 if the model’s rollout $o_i^t$ matches the ground truth $a$, and 0 otherwise.
> The term $e(o_i^t)$ is the average token-wise entropy of rollout $o_i^t$, $e(\overline{o_i^{t+}})$ is the mean entropy across all positive rollouts, and $\lambda$ is a threshold used to filter out high-entropy negative rollouts.
>
> + **Step 2: Compute Rollout-level Explorability**: We then combine the entropy $e(o_i^t)$ by the absolute value of the advantage $|\hat{A_i}|$  and $I(q, a, o_i^t)$ to compute the explorability for a single rollout $o_i^t$ as:
> $E(q,a,o^t_i) = |\hat{A_i}| \cdot e(o_i^t) \cdot I(q, a, o_i^t)$.
> This ensures that samples with both high **exploration potential** (entropy) and **appropriate difficulty** (absolute advantage) are prioritized.
>
> + **Step3: Sample-level Aggregation**: Finally, to evaluate a sample's exploration potential over time, we aggregate rollout-level explorability across the group over a sliding window of recent epochs to obtain sample-level explorability $\mathcal{E}$.
> $\mathcal{E}(q,a,\{O^t\}_{t=e-s+1}^e)=$
>
> $\frac{1}{s} \sum_{t=e-s+1}^{e} \frac{1}{G} \sum_{i=1}^G E(q,a,o^t_i)$.
> Here, $s$ is the sliding window size of recent epochs, and $G$ is the number of rollouts per sample at a step.
>
> To improve the readability of our paper, we have incorporated this detailed explanation into **Section 3.3.1**.
>
> ### **Q4: In equation (6), what does top-α_e% means, which term should be the top of which set? The same for "bottom-ρ%"**
>
> In Equation (6):
> + **top-$\alpha_e\%$** refers to the top $\alpha_e\%$ of samples in the entire batch $\mathcal{B}$, ranked by their explorability $\mathcal{E}$ (i.e., those with the **highest explorability scores**).
> + **bottom-$\rho\%$** refers to the bottom $\rho\%$ of samples in the entire batch $\mathcal{B}$, ranked by their **historical exploration frequency** (i.e., those that have been selected the fewest number of times up to the current epoch $e$).
>
> To improve readability, we have incorporated this clarification into **Section 3.3.2**.
>
> ### **Q5: In the final objective function (5), the reviewer didn't see the role of the final offline dataset D^{sub} if the reviewer didn't miss something. So what are the roles of the offline and online phase?**
>
> We thank the reviewer for catching this inconsistency.
> The original Equation (5) contained a mistake, which we have now corrected in the revised version.
> Here is the intended pipeline:
> + **Offline Phase**: Our multi-dimensional curation strategy is applied to the full offline dataset $\mathcal{D}$ to select a high-quality subset $\mathcal{D}^{\text{sub}}$.
> + **Online Phase**: The online data selection operates on this curated subset $\mathcal{D}^{sub}$, not on the original $\mathcal{D}$.
>
> Accordingly, we have corrected **Equation (5)** from the erroneous $\mathcal{B} \sim \mathcal{D}$ to the accurate $\mathcal{B} \sim \mathcal{D}^{\text{sub}}$.

---

> ### Author Response · Authors · 2025-11-19
> **Rebuttal by Authors (3/3)**
>
> ### **Q6: Could the author explain more about this claim/intuition around line 177: "To promote diversity, we use Determinantal Point Process (DPP) (Kulesza & Taskar, 2012a) to identify a subset that maximizes the determinant of the corresponding similarity submatrix: maxY ⊆P det(Y ), where Y is the submatrix of the full similarity matrix P, and det(·) represents its determinant value. This refers to selecting samples that form a larger volume in the feature space, with a larger volume indicating greater diversity." Why a larger determinant of the similarity matrix P may lead to diversity? Since intuitively, if some entries in P_ij is small, that means the corresponding sample i and j are different, giving more diversity. So should we just find those regions with small elements in P? Why the determinant is related to the diversity of samples?**
>
> Thank you for raising this question regarding the connection between determinant maximization and diversity. We are happy to provide further clarification.
>
> **Simple Case: Two Samples**: Consider selecting two samples. The subset similarity matrix is:
>
> \begin{equation}
> \mathbf{Y} = \begin{bmatrix}
> 1 & p_{12} | \\
> p_{12} & 1
> \end{bmatrix}
> \end{equation}
>
>
> The determinant is $1-p_{12}^2$. Maximizing diversity here corresponds to minimizing the pairwise similarity $p_{12}$, which is achieved precisely by maximizing the determinant.
>
> **General Case: Multiple Samples**: For a subset of $m$ samples, the similarity submatrix $Y$ has ones on the diagonal and pairwise similarities off the diagonal.
> The determinant $\det(\mathbf{Y})$ measures the volume spanned by their feature vectors in the reproducing kernel Hilbert space. This volume satisfies:
> $Vol_m(x_1, \dots, x_m) = \prod_{k=1}^m |x_k| \cdot \prod_{k=2}^m \sin\phi_k$.
> where each $\sin\phi_k \in [0,1]$ quantifies the orthogonality of $\mathbf{x}_k$ to the span of previous vectors.
> + $\det(Y)$ is maximized when all $\sin\phi_k = 1$ (i.e., **when the vectors are mutually orthogonal**, indicating high diversity).
> + $\det(Y)$ is zero when any $\sin\phi_k = 0$, **indicating linear dependence** and minimal diversity.
>
> Thus, maximizing $\det(\mathbf{Y})$ directly promotes a diverse set of samples.
>
> We have included a more rigorous explanation and proof in **Appendix D.2** to further support this point.
>
> **Determinant vs. Individual Similarities**: While small individual $P_{ij}$ values do indicate pairwise diversity, they **do not ensure global diversity** for the entire set.
> Simply minimizing individual off-diagonal entries does not guarantee the diversity of the whole set, it may still contain clusters or redundant elements.
> In contrast, the determinant accounts for all pairwise similarities simultaneously and ensures the selected set is jointly diverse.

---

> ### Comment · Reviewer_L77o · 2025-11-26
> **Response to the author**
>
> Thank you for your detailed response and revisions. I have reviewed the changes, and most of my comments have been addressed.
>
> One more question:
> The author mentioned that it is due to the combination of both offline and online data curation. But it seems that it is also not a combination of prior works that focus on offline or online. So it remains curious what are the key contribution that make the data efficiency happen or both terms matters. The reviewer know there is ablation study. But more explaination will be helpful.

---

> > ### Author Response · Authors · 2025-11-26
> > **Further Response to Reviewer L77o**
> >
> > Dear Reviewer L77o,
> >
> > Thank you very much for taking the time to review our response.
> > We are glad that most of your concerns have been addressed.
> > We particularly appreciate your **constructive suggestions regarding the methodology section**, which have been incorporated into our revised version and have significantly improved the paper's logic and readability.
> >
> > ### **Q1: The author mentioned that it is due to the combination of both offline and online data curation. But it seems that it is also not a combination of prior works that focus on offline or online. So it remains curious what are the key contribution that make the data efficiency happen or both terms matters. The reviewer know there is ablation study. But more explaination will be helpful.**
> >
> >
> > Thank you for this thoughtful question.
> > We appreciate the opportunity to further clarify how both offline and online data curation contribute to data efficiency.
> >
> > We believe that **both** components work together to improve data efficiency:
> >
> > + **In the offline phase**, the RLVR dataset is particularly large as it requires only questions and answers without intermediate reasoning steps. Therefore, our main goal here is selecting samples that are both **diverse and influential**, which helps **filter out redundant data**. Additionally, sample difficulty is an important factor in RLVR, because **samples with moderate difficulty** tend to accelerate model improvement more effectively. Thus, in the second stage, we use offline rollouts to select a subset that best matches the model’s current capabilities for RLVR training.
> > + **In the online phase**, samples with **strong exploration potential** can guide the model to perform higher-quality exploration and generate better rollouts. Our proposed **explorability metric**, which combines entropy and difficulty, helps select the most valuable data during online training.  Moreover, we **dynamically replay under-explored samples** that may become more valuable in later training stages, leading to better final convergence.
> >
> > Thank you again for your valuable time and feedback. If you have any further questions, we would be very grateful for the opportunity to discuss them further.
> >
> > Best regards,
> >
> > Authors

---

### Author Response · Authors · 2025-11-24
**General Response**

Dear Area Chairs and Reviewers,

We sincerely thank you for your time, effort, and constructive feedback on our work.
Our paper focuses on improving **data efficiency** in Reinforcement Learning with Verifiable Feedback (RLVR), introducing **optimized strategies** for both offline and online data selection.
We are truly encouraged that the reviewers acknowledged the contributions of our work, which we briefly summarize below:

+ **Novelty**: **``Reviewers Gqob and 1LYZ``** highlighted the novelty of our approach—introducing a comprehensive RLVR learning paradigm that **systematically integrates offline data curation and online data selection**.
+ **Well-Designed Framework**: Our proposed multi-dimensional offline data curation and explorability-guided rollout pruning were recognized as **refined and logically compelling** (**``Reviewers Gqob and 1LYZ``**).
+ **Comprehensive Experiments**: **``All reviewers``** appreciated the **extensive** experiments—covering different LLMs, benchmarks, dataset sizes, sampling, and rollout ratios—which validate the effectiveness, efficiency, and robustness of our method.
+ **Practical Impact**: **``Reviewer Gqob``** noted that our work “makes a substantial **practical contribution** that could lower the barrier to entry for RLVR training.”

We appreciate the insightful comments and suggestions from the reviewers and provide additional experimental results, clarifications, and manuscript revisions during the rebuttal phase.

**``Reviewer L77o``**: the corresponding revisions are highlighted in **light yellow**.

**``Reviewer Gqob``**: the corresponding revisions are highlighted in **light green**.

**``Reviewer 1LYZ``**: the corresponding revisions are highlighted in **light blue**.

Below, we summarize the main updates.

+ **Expanded discussion of related work**: included a new **Table 1** for comparison (**``Reviewer L77o``**), added related work in **Section 2** (**``Reviewer L77o``**) and summarized our contributions in the **Introduction Section** (**``Reviewer Gqob``**).
+ **Improved writing**: enhanced the methodology description in Section 3.3.1 and provided more detailed variable descriptions in Section 3.3.2, along with correcting an error in Equation (7) (**``Reviewer L77o``**).
+ **Experiments**:
    + **Detailed Efficiency Analysis**: added comprehensive efficiency results in **Table 2** (**``Reviewer 1LYZ``**)
    + **Comprehensive Ablation Study**: conducted ablation study on explorability measurement components in **Table 4** with corresponding explanations in **Section 3.3.1** (**``Reviewer Gqob``**).
    + **Additional Baselines**: incorporated easy-to-difficult curriculum strategy in **Figure 6(a)** (**``Reviewer Gqob``**)
    + **Extended Discussion**: Added analysis of dynamic offline data curation strategy in **Appendix F.3** (**``Reviewer 1LYZ``**)
+ **Theoretical Enhancement**: Included discussion on why determinant can measure sample diversity in **Appendix D.2**. (**``Reviewer L77o``**)


We believe these revisions have significantly improved the technical rigor, clarity, and completeness of our work, and we have incorporated them into the revised version.

Please do feel free to let us know if you have any further questions.
We are fully committed to addressing them and welcome the opportunity for further discussion.

Best Regards,

Authors

---

### Comment · Area_Chair_GS9x · 2025-11-25

Dear reviewers:

The authors have submitted their rebuttal, and we now require your follow-up assessments to move the decision process forward. Please review the authors’ responses and update your evaluations accordingly.

Your prompt follow-up is necessary for us to finalize the meta-review.

Kindly submit your updates as soon as possible.

Best,

Area Chair

---

### Author Response · Authors · 2025-11-29
**Summary of Our Responses and Revisions during the Rebuttal Period**

Dear Area Chairs,

We sincerely thank you for your time and effort in handling our paper.

Our work focuses on improving **data efficiency** in Reinforcement Learning with Verifiable Feedback (RLVR), proposing **optimized strategies** for both offline and online data selection.

---

## **1. Reviewer Feedback**

During the rebuttal period, our responses and revisions were **positively acknowledged by all reviewers**:
+ **``Review L77o``** acknowledged that **most of their initial concerns were addressed** and raised one new question. We responded to this point during the rebuttal period, but regret that we are unable to continue the discussion further since reviewers can not comment now. **``(4, no further discussion)``**
+ **``Review Gqob``** appreciated our detailed responses and the additional experiments conducted during rebuttal, and decided to **raise their rating from 4 to 6 and confidence from 2 to 4**. **``(4->6)``**
+ **``Review 1LYZ``** also confirmed that **their main concerns were resolved** and raised two new questions. But it is a pity that the reviewers could not review our detailed clarifications and continue the discussion. **``(6, no further discussion)``**

---

## **2. Our Strengths**

We are truly encouraged that the reviewers acknowledged the contributions and strengths of our work, which we briefly summarize below:

+ **Novelty**: **``Reviewers Gqob and 1LYZ``** highlighted the novelty of our approach—introducing a comprehensive RLVR learning paradigm that **systematically integrates offline data curation and online data selection**.
+ **Well-Designed Framework**: Our proposed multi-dimensional offline data curation and explorability-guided rollout pruning were recognized as **refined and logically compelling** (**``Reviewers Gqob and 1LYZ``**).
+ **Comprehensive Experiments**: **``All reviewers``** appreciated the **extensive** experiments—covering different LLMs, benchmarks, dataset sizes, sampling and rollout ratios—which validate the effectiveness, efficiency, and robustness of our method.
+ **Practical Impact**: **``Reviewer Gqob``** noted that our work “makes a substantial **practical contribution** that could lower the barrier to entry for RLVR training.”

---

## **3. Main Responses and Revisions**

Besides, we appreciate the insightful comments and suggestions from the reviewers, and provide additional experimental results, clarifications, and manuscript revisions during the rebuttal phase.
Below we summarize our responses and revisions:

+ **Expanded discussion of related work**: included a new **Table 1** for comparison (**``Reviewer L77o``**), added related work in **Section 2** (**``Reviewer L77o``**) and summarized our contributions in the **Introduction Section** (**``Reviewer Gqob``**).
+ **Improved writing**: enhanced the methodology description in **Section 3.3.1** and provided more detailed variable descriptions in **Section 3.3.2**, and corrected an error in **Equation (7)** (**``Reviewer L77o``**).
+ **Experiments**:
    + **Statistical Validation**: provided standard deviations and significance tests in **Table 3** (**``Reviewer Gqob``**)
    + **Detailed Efficiency Analysis**: added comprehensive efficiency results in **Table 2** (**``Reviewer 1LYZ``**)
    + **Comprehensive Ablation Study**: conducted an ablation study on explorability measurement components in **Table 4**, with corresponding explanations in **Section 3.3.1** (**``Reviewer Gqob``**).
    + **Additional Baselines**: incorporated an easy-to-difficult curriculum strategy in **Figure 6(a)**. (**``Reviewer Gqob``**)
    + **Extended Discussion**: Added analysis of dynamic offline data curation strategy in **Appendix F.3** (**``Reviewer 1LYZ``**), and further discussed progressively reducing the dataset size across resampling phases in our response to **Q2 of the follow-up questions** posed by **``Reviewer ILYZ``**.
+ **Theoretical Enhancement**: Added discussion on why the determinant can measure sample diversity in **Appendix D.2**. (**``Reviewer L77o``**)
+ **Clarification Advantages of Our Method**: Explained the benefits of our two-step offline data curation in response to **Q4** to **``Reviewer Gqob``**; clarified how our method improves data efficiency in our response to **follow-up questions** proposed by **``Reviewer L77o``**; and emphasized that the offline phase requires minimal time in our response to **Q1 of the follow-up questions** posed by **``Reviewer ILYZ``**.

---

We believe these revisions have strengthened the technical rigor, clarity, and completeness of our work.
**All changes have been incorporated into the revised version, with modifications clearly highlighted**.

Thank you once again for your valuable time and consideration.

Best Regards,

Authors

---

### Meta-Review · Area_Chair_bMXM · 2026-01-06

**Summary:**

In this paper, the authors considered the data curation problem for LLM reasonging capabability improvement both in offline and online perspectives. In the offline case, the authors selected a subset of data from a large corpus based on the determinant of the selected data embedding vetors, capturing the volume spanned by these vectors for diversity, and page rank algorithms to catch important samples. In the online case, they considered pruning of rollouts to maixmized explorability of a rollout, defined as the product of advantage, average token entropy and some exclusion factor.  They showed their method yields significant speed up in training time.

**Reviewer Concerns:**

The reviewers raised the following concerns:
- Writing should be improved. Too many components are in this paper and they should be well organized and explained.
- Statititical validation of experimental results is required.
- Thorough search of related works is required.
- Several components such as DPP and the page rank algorithm are well known.

AC comment:  If the selected subset size is much larger than the dimension of embedding space, then the determinant is likely to be zero almost surely. So, this method works for LLMs with large embedding space dimensions so that it can contain the linear subspace spanned by the data subset embedding vectors. Another concern is that the computational complexity should be thoroughly analyzed. For the offline data curation steps, computing the determinants of many combinations and computing accuracy of each output sequence of a group for the current model for sampling may incur significant amount of computation.

**Reviewer Scores:**

The original review scores are 6 (Reviewer 1LYZ), 4 (Reviewer L77o), 4 (Reviewer Gqob). It is expected that Reviewers L77o and Gqob with orginal score 4 would raise their scores to 6, from their discussion.

Indeed, this paper presents a meaningful way to select a non-redundant and important data subset from a large corpus for LLM post-training. Although DPP and PageRank algorithms are well known, their application to LLM data selection is useful, and applying the PageRank algorithm seems nice and unexpected. Furthermore, sampling questions with medium difficulty w.r.t. the current model well addresses the drawback of GRPO, which is based on the relativeness of rewards within group. That is, if all actions (outputs) within a group for a question solve the question, then all rewards in the group are 1; the mean is one; the delta, i.e., ( r i - mean(r i)), is all 0, and hence, theoretically, all advantages are zero for all actions. On the other hand, if all actions within a group fail. then again all deltas for the group is zero. In both cases, the standard deviation also reduces to zero. We may have a 0/0 situation. So, for GRPO, we need half of actions solve the problem and the other half fail the problem for good advantage extraction. Authors well implemented this idea for question sampling. So, AC recommends acceptance of this work considering the importance of data efficiency in LLM training and authors' thorough responses.

---

### Decision · Program_Chairs · 2026-01-26

Accept (Poster)